# Comparison and Verification of Reliability Assessment Techniques for Fuel Cell-Based Hybrid Power System for Ships

**Hyeonmin Jeon** [1] **, Kido Park** [1,2] **and Jongsu Kim** [1,*]

[1]  Department of Marine System Engineering, Korea Maritime and Ocean University, Busan 49112, Korea; jhm861104@kmou.ac.kr (H.J.); kdpark@krs.co.kr (K.P.)
[2]  Future Technology Research Team, Korea Resister, Busan 46762, Korea
*  Correspondence: jongskim@kmou.ac.kr; Tel.: +82-(0)102-558-1197

**Abstract:** In order to secure the safe operation of the ship, it is crucial to closely examine the suitability from the design stage of the ship, and to set up a preliminary review and countermeasures for failures and defects that may occur during the construction process. In shipyards, the failure mode and effects analysis (FMEA) evaluation method using risk priority number (RPN) is used in the shipbuilding process. In the case of the conventional RPN method, evaluation items and criteria are ambiguous, and subjective factors such as evaluator's experience and understanding of the system operate a lot on the same contents, resulting in differences in evaluation results. Therefore, this study aims to evaluate the safety and reliability for ship application of the reliability-enhanced fuel cell-based hybrid power system by applying the re-established FMEA technique. Experts formed an FMEA team to redefine reliable assessment criteria for the RPN assessment factors severity (S), occurrence (O), and detection (D). Analyze potential failures of each function of the molten-carbonate fuel cell (MCFC) system, battery system, and diesel engine components of the fuel cell-based hybrid power system set as evaluation targets to redefine the evaluation criteria, and the evaluation criteria were derived by identifying the effects of potential failures. In order to confirm the reliability of the derived criteria, the reliability of individual evaluation items was verified by using the significance probability used in statistics and the coincidence coefficient of Kendall. The evaluation was conducted to the external evaluators using the reestablished evaluation criteria. As a result of analyzing the correspondence according to the results of the evaluation items, the severity was 0.906, the incidence 0.844, and the detection degree 0.861. Improved agreement was obtained, which is a significant result to confirm the reliability of the reestablished evaluation results.

**Keywords:** hybrid power system; failure mode and effects analysis; risk priority number; ship safety; Kendall's coefficient

## 1. Introduction

In the early 2000s, the Maritime Safety Committee (MSC) of the International Maritime Organization (IMO) adopted the item goal-based new ship construction standards (GBS) [1], which present new ship design and construction concepts for the long-term organizational work plan. They then developed safety level approach (SLA)-based GBS that are applicable to all ships [2]. The IMO has since actively strengthened the Safety of Life at Sea (SOLAS) standards based on the GBS to reduce the underlying causes of marine accidents and environmental pollution from ship construction and to prioritize ship safety [3].

To assess safety in the ship construction stage, a hazard identification and risk analysis (HIRA) is conducted to identify and evaluate the risk of the system installed in a ship. Specific evaluation

methods for analyzing hazards in HIRA include hazard identification (HAZID), hazardous operability (HAZOP), what-if/checklist, and failure mode and effects analysis (FMEA) [4].

FMEA, a type of risk assessment method, was developed for the Apollo project by the National Aeronautics and Space Administration (NASA) in the mid-1960s. Since then, three major US automakers have introduced their own assessment system "QS-9000" [4]. However, FMEA is the most common way to evaluate device reliability [5]. It is a preventive reliability assessment method performed at the design stage for system or component changes, and it uses an empirical perspective for the analysis and component changes to achieve the optimal results. It is extensively used to assess the design, process, and system risks across all industries including the shipbuilding and marine sectors.

FMEA is advantageous in that it enables systematic analysis using simple methods. The evaluation criteria for the expected severity, occurrence, and detection are established using the risk priority number (RPN) technique, and the failures for individual components are assessed [5,6]. These results are combined to obtain the criticality. However, the logic is inferior to other methods because it uses a qualitative evaluation, and the evaluation results may vary depending on the experience or inclination of the evaluator assessing the failure.

Researchers have performed various studies to increase the objectivity of FMEA. Research has been conducted on an approach combining FMEA and the Boolean representation method (BRM) [7], a method that describes the elements required for FMEA and then develops and applies an appropriate FMEA form for an effective evaluation. Studies have also used a computer system method that supports FMEA evaluations on the Internet [8], the risk priority ranks (RPR) approach to prioritize failure modes [9], a method based on fuzzy logic that considers the interdependence between various failure modes [10–14], a fuzzy-based FMEA performance improvement method using GRAY relationship theory [15], and a method that provides a framework for automatically generating FMEA from past FMEA data using functional inference techniques [16]. Research has additionally been conducted on how to most effectively apply the FEMA system due to difficulties related to its numerous subsystems and the lack of consideration for the indirect relationship between the components in the RPN technique.

In particular, in recent years, in order to apply environmentally friendly ships, ships using hybrid fuel cells, batteries, etc. are being operated mainly on small coastal ships. These vessel systems are very different from the diesel engines used as conventional ship power sources, so new FMEA evaluation criteria and items should be provided to evaluate the safety and reliability of such vessels. However, even in shipyards that are currently building vessels, FMEA evaluation criteria or items have not been specifically set.

Therefore, in this study, the proposed FMEA was conducted to secure the safety and reliability for applying the fuel cell-based (molten-carbonate fuel cell (MCFC; 100 kW), battery (30 kW), and diesel generator (50 kW)) test bed to the actual ship. We analyzed various problems in evaluating RPN, which is mainly used in FMEA, and formed an FMEA expert team to select evaluation criteria and items. As a result, we developed a worksheet applying the reestablished RPN evaluation criteria, and applied Kendall's coefficient of correspondence to the existing evaluation results and the reestablished evaluation results for objective determination of the reestablished evaluation criteria. It was confirmed that the reestablished assessment in the FMEA evaluation of the combined power source showed more reliable results. In addition, the criteria for establishing countermeasures based on the results of the FMEA were prepared, and the proposed evaluation method was found to be effective for the application of the assessment of the safety and reliability of the combined power source.

## 2. Theoretical Background of FMEA and RPN Introduction

### 2.1. What is FMEA?

FMEA was first used in the NASA Apollo project in the 1960s. In 1974, it was used to develop United States Navy missiles and was established as the United States MIL-STD-1629 standard. Afterwards, the QS-9000 standard was established by the United States automobile industry, and FMEA was introduced in all industries, including shipbuilding [4]. The FMEA method prioritizes resources,

ranks risks, and creates an activity and control plan to analyze the target system [5,6], thereby analyzing failure types and their influence and examining improvement measures with consideration of criticality [17].

The objectives of FMEA are as follows:

(1)   Identify potential defects inherent in the system and evaluate the severity of their effects.
(2)   Identify key management items.
(3)   Recognize important potential design and process defects
(4)   Prevent severe product accidents and customer complaints.
(5)   Provide a basis for establishing sector-specific measures to eliminate or reduce defects.
(6)   Enhance efficiency by verifying design and production problems.

Figure 1 illustrates the typical FMEA process.

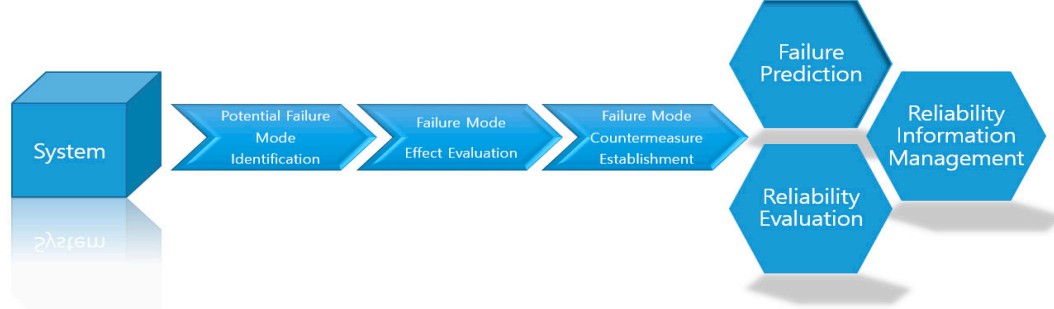

**Figure 1.** Block diagram of the typical failure mode and effects analysis (FMEA) process.

*2.2. RPN Technique*

RPN is a relative potential failure evaluation measure that is primarily used in FMEA, and it prioritizes management and corrective actions. RPN consists of three items: severity (S), occurrence (O), and detection (D). The value of each item is divided into 10 levels from 1 (bad) to 10 (good); the value of RPN is between 1 and 1000 and is obtained by the product of each item [17–20]:

$$RPN = S \times O \times D.$$

Figure 2 shows the meaning of each RPN item. S affects the customer in relation to the process or product when a potential failure occurs. The degree of S of the effect is scored and evaluated, and a reduction in the S class can only be affected through design changes. O refers to the possibility of the cause and mechanism taking place. The O class must be consistent. D indicates whether a potential failure mode and its cause can be discovered or detected.

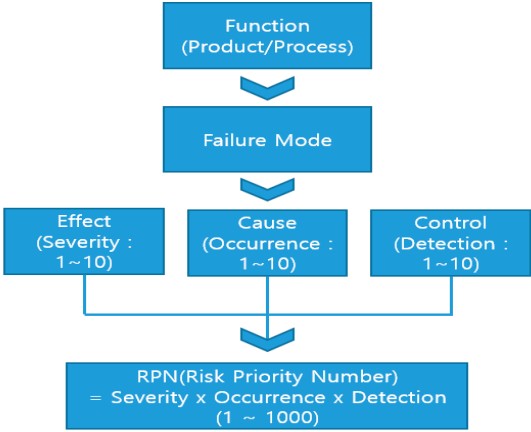

**Figure 2.** The meaning of each risk priority number (RPN) item.

## 3. Problem Analysis and Solution of the Existing RPN Evaluation Method

*3.1. Problems with the Existing RPN Evaluation Method*

There are numerous problems with the existing RPN evaluation method; the following issues directly affect the evaluation [4,18,19].

(1)　S, O, and D, the components of RPN evaluation, are influenced by many subjective factors that depend on the evaluator. Therefore, if the evaluator is insufficiently experienced with and knowledgeable of the system, the results may differ from those of another evaluator using the same criteria. The evaluation results of RPN are sensitive to the score variations of each component (S, O, and D). Therefore, if the evaluation criteria are unclear, the evaluation results can differ. For example, assuming that S and O are fixed at a class of 7 and D has a 1 class difference, the RPN score varies by a sizeable 64 points.

(2)　In some cases, the evaluation criteria are inappropriate for the particular product or system being evaluated. For example, the RPN standards for shipbuilding differ significantly from those of automakers; applying uniform criteria to both systems greatly increase the likelihood of issues occurring when operating the product.

(3)　While the evaluation components of RPN can be assessed individually, the influence of S, O, and D on each other is not taken into account. For example, assume that for RPN1, S, O, and D are 4, 5, and 6, respectively, and the RPN has value of 120. The S, O, and D of RPN2 are 4, 6, and 6, respectively, and the total RPN is 144.

(4)　The evaluator responsible for the system is in charge of establishing and implementing measures; therefore, they may be reluctant to thoroughly evaluate the system RPN and may intentionally underestimate it. RPN underestimation and product recalls can lead to enormous time and financial losses, and damage to the manufacturer's image.

(5)　If the system evaluation criteria are ambiguous, the evaluator may assess them arbitrarily, leading to vast RPN differences between evaluators.

Overestimating RPN leads to the implementation of unnecessary countermeasures and an excessively safe system design, increasing system installation costs. In contrast, if RPN is underestimated, the appropriate measures for the effects of each failure mode are not established, risking the preventability of future accidents. This can then lead to huge time and money losses. For example, in 1998, GM in the United States received a $4.97 billion fine to compensate the explosion of an automobile fuel tank following a traffic accident. According to the company internal report, the reliability assessment recognized that there was an explosion risk if the fuel tank was manufactured at a low cost. In spite of having access to this information, the vehicle was released without any modifications, leading to the highest payout for individuals in American history [5].

Although FMEA poses numerous problems, it is the most frequently applied reliability evaluation method across all industries because of its simple and systematic analysis. To strengthen the FMEA evaluation performance to supplement the existing problems of FMEA, researchers have investigated methods and approaches from various perspectives [21,22], including a method where, after pre-selecting the factors necessary for FMEA [23], the relationship between the failure mode and effect can be determined by applying various control methods such as fuzzy logic, neural network, functional inference theory [10–14,24–28]; a FMEA matrix, which graphically assesses the relationship between the elements of a system, failure modes, and failure effects [29,30]; methods to effectively prepare the appropriate FMEA form for a given objective [31,32]; methods to provide a worksheet that automatically generates the FMEA using past FMEA data [33]; and other approaches to derive more objective FMEA results [34].

### 3.2. Improvement in the RPN Technique and Improvement of the Evaluation Method Using Kendall's Concordance Coefficient

To improve the problems that occur in RPN evaluation using FMEA and derive objective results, as shown in Figure 3b, this study precisely identified the potential failure types matching the characteristics of the fuel cell-based hybrid power system for ships and analyzed the RPN evaluation criteria. Figure 3a shows the process for determining the existing RPN evaluation items, and Figure 3b shows the process for determining the RPN evaluation items applied in this study.

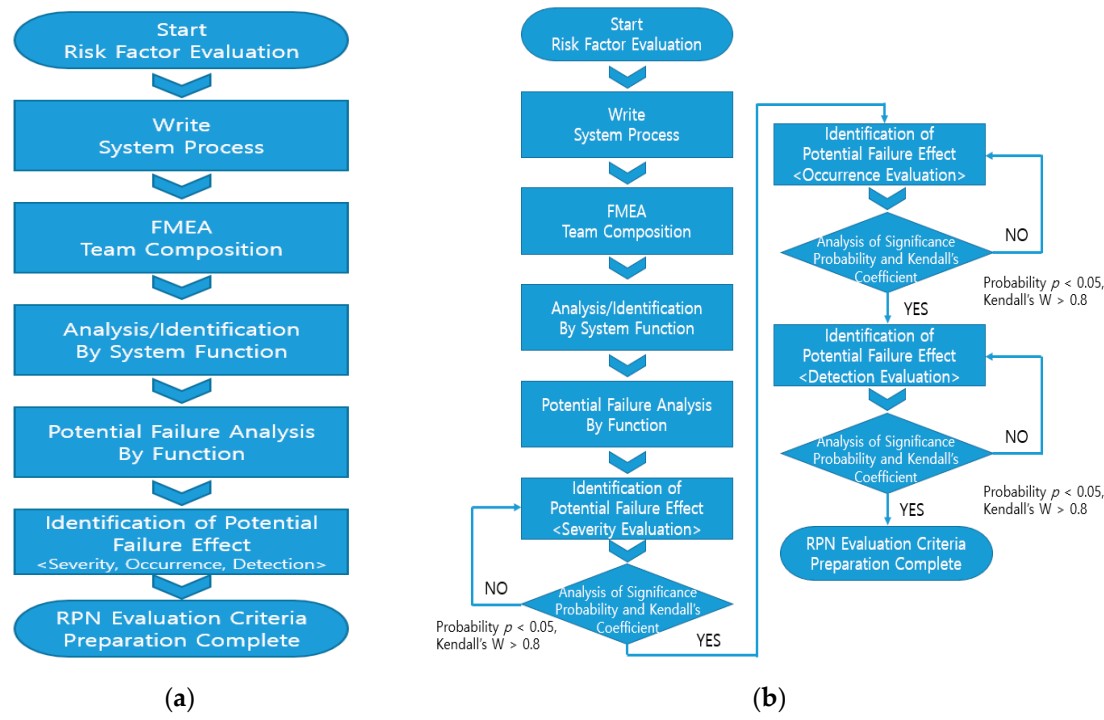

**Figure 3.** (**a**). The process for determining the existing RPN evaluation items. (**b**). The process for determining the RPN evaluation items applied in this study.

Minimizing differences between the results of various evaluators can increase RPN evaluation reliability. To increase the reliability of the evaluation results, team members with a certain amount of experience in specialty fields were selected for the FMEA team in this study. They performed a system analysis by function. The FMEA team is aware of the problems with existing FMEA because it has been working in the field for a certain period of time and selected experts with basic experience in FMEA evaluation. Therefore, we understand the importance of FMEA evaluation criteria and item setting.

The composition of the FMEA team and the criteria for selecting experts are as follows:

(1) The FMEA team consists of 10 experts for the group;
(2) The selected experts are currently employed in shipyards, research institutes, classification society, engine makers, test and certification institutes, and educational institutions;
(3) Over 5 years of experience in fuel cell, battery, and diesel engine system;
(4) Have more than 10 times of experiences in evaluation FMEA.

Based on the functional analysis of potential failures, this study designed clear evaluation criteria for S, O, and D. The existing effects of potential failures were identified, then the RPN evaluation criteria were created, and an evaluation was immediately performed. However, when creating the RPN evaluation criteria, this study identified the effects of the potential failures of S, O, and D. The reliability of the evaluation criteria were then confirmed, and the criteria were established using the following procedure.

First, the evaluation items for S, O, and D were established, after which the following research hypothesis for the evaluation items was set: "the evaluation scores by item of the evaluators will be similar.". The FMEA team then performed its own internal evaluation, confirming the significance probability results for the reestablished evaluation items and validating the research hypothesis. Next, the team RPN internal evaluation results were compared with Kendall's concordance coefficient to determine the reliability of each evaluation item. In this paper, Kendall's coefficient of consensus mentioned to verify the reliability of the evaluation items is one of the methods used in nonparametric statistics to analyze the relationship between phenomena measured on the sequence scale [35]. Kendall's coincidence coefficient is typically used for attribute agreement analysis, with coefficient values ranging from 0 to 1. The higher the value of the coefficient, the stronger the association. If the coefficient is greater than 0.9, the relevance is considered very high and the high or significant Kendall's coefficient means that the evaluators apply essentially the same standard when evaluating the sample [36]. Applying the same criteria decreases the ambiguity of the evaluation items, removing arbitrariness and encouraging objectivity. Then, the significance probability of the evaluation criteria items and the results of Kendall's concordance coefficient were determined. If the reliability of the evaluation criteria was lower than the threshold, then the process returned to the previous steps to identify the effect of potential failures; once the reliability of the evaluation criteria reached the threshold, the evaluation criteria was confirmed.

This final evaluation criteria were then used as the basis to assess the external evaluators. Finally, by comparing the results with the existing evaluation criteria, this study numerically confirmed the high reliability of the reestablished evaluation criteria.

## 4. FMEA Methodology of This Study

### 4.1. FMEA Procedure of This Study

According to the IEC 60812 standard, the FMEA procedure can be divided into three steps: the preparation, performance, and finishing [37].

#### 4.1.1. Preparation Step

To implement FMEA, it is necessary to examine the criteria applied to each power source and hybrid power system configured in the test bed. As a marine fuel cell was applied to an Eidesvik Offshore support vessel of, this study collected and referenced safety-related data such as fuel supply facilities, fire protection facilities, and ventilation systems. This study also examined the "Guidance for Fuel Cell Systems on Board of Ships" published in the Korean Register of Shipping, the "Approval in principle fuel cell installation for LNG Tanker" standard, and the "Guideline for the use of fuel cell systems on board of ships and boats" published in the Det Norske Veritas-Germanischer Lloyd (DNV-GL) registrar [38].

The FMEA worksheet, an important component of FMEA, should be confirmed before performing FMEA. S classification, one of the items in the worksheet, is particularly important; this should be completed with reference to Table 1, which indicates the severity class presented in IMCA M 166 [39].

**Table 1.** Example of severity ratings as outlined in IMCA M 166.

| Classification | Degree | Description |
| --- | --- | --- |
| 1 | Minor | Functional failure of machinery and process components without the effects of injury, damage, or contamination. |
| 2 | Critical | Failure without severe damage, contamination, or injury to the system. |
| 3 | Major | Critical damage to the system, including the possibility of injury or minor contamination. |
| 4 | Catastrophic | Failure causing total system loss with high possibility of fatal injury or large contamination. |

### 4.1.2. Performance Step

In the FMEA performance step, the causes, effects, countermeasures, and severity for each failure mode were discussed; these items were recorded and organized through a worksheet [4]. Here, the effect of the failure mode could be confirmed through the experience of the evaluator, drawings, or simulations. The RPN was used in the evaluation, which indicates the S, O, and D when performing FMEA [5].

### 4.1.3. Finishing Step

In the FMEA finishing step, FMEA was performed, and all the generated data were organized into a report. The standards, design drawings, single line diagrams, and worksheets used in the report should be organized in a manner that is useful as design data and for the revision step of the system conducted later on. Figure 4 is the FMEA one cycle.

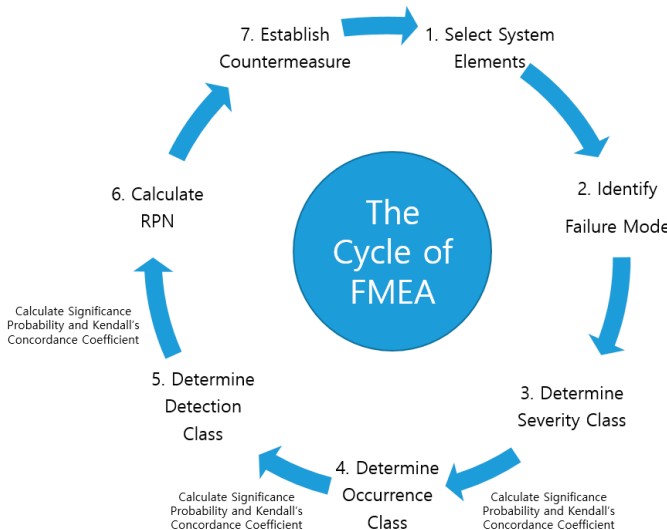

**Figure 4.** The FMEA one cycle.

*4.2. RPN Evaluation Criteria Reestablished in This Study*

### 4.2.1. RPN Evaluation—Severity Criteria

Table 2 shows the S criteria, one of the RPN evaluation factors for a fuel cell-based hybrid power system. The newly applied evaluation criteria were classified as 1, 2, or 3 to enable the accurate evaluation of S from the system and the customer perspectives. Evaluation Criteria 1 simultaneously reflects both the system and customer effects, while Evaluation Criteria 2 contains the corresponding more detailed effects. Evaluation Criteria 3 consists of the effects on the development stage.

**Table 2.** RPN criteria for severity.

| Class | Severity | | | |
|---|---|---|---|---|
| | Evaluation Criteria 1 | | Evaluation Criteria 2 | Evaluation Criteria 3 |
| | System Effects | Customer Effects | Detailed Effects | Development Effects |
| 10 | With no prior warning, system operations are affected or there are inconsistent international regulations. | There is a major failure related to safety (casualty), such as ignition or explosion without prior warning, posing a risk to customers. | There is a gradual failure after a potential failure related to casualties. | The system development task is dropped. |
| 9 | Even with prior warning, system operations are affected or there are inconsistent international regulations. | Even with prior warning, there is a major failure related to safety (casualty), such as ignition or explosion, posing a risk to customers. | There is the sudden occurrence of a dangerous failure directly related to a casualty; items are regulated by the government. | Product concept is changed. |
| 8 | The system fails to operate due to the loss of major system functions. | Customers are very dissatisfied, the product does not function, and the product must be disposed of. | Equipment is damaged, it does not operate correctly, and it must be completely disposed of. | There is a change in the assembly component design (customer specifications (spec) out). |
| 7 | The system can operate, but the product malfunctions. | Customers are dissatisfied, and the product does not work properly. | While system rework or repair is possible, its functionality has already been severely affected, and selective disposal is required. | There is a change in the component design concept (insufficient customer spec in margin). |
| 6 | The main functionality of the system operates normally, but the peripherals are inoperable due to performance deterioration. | Customers are slightly dissatisfied, and simple repair and rework is needed. | Product functionality is affected, and customers are dissatisfied; a partial simple repair is required, and total rework is possible. | There is a change in the component design (internal spec out). |
| 5 | The main functionality of the system operates normally, but the peripherals do not operate properly due to performance deterioration. | | A repair is required due to degraded product functionality; some functions do not work, and selective rework is possible. | The component is optimized (insufficient spec in, margin). |
| 4 | When the system is manufactured, certain peripheral functions are degraded because finishing was not performed properly. | At least half of the customers are mildly dissatisfied; functionality is somewhat affected, but no repair is required. | There is a weak effect on product operations; customers feel discomfort. | Process matching occurs (insufficient spec in, margin). |
| 3 | | Some of the customers are mildly dissatisfied; functionality is somewhat affected, but no repair is required. | The output or functionality of the unit process is slightly degraded. | There is a slight effect on product characteristics (spec irrelevant). |
| 2 | There is almost no effect on the system. | There is almost no effect on customers (next process), and there are no quality defects. | There are no effects on the system, product functionality, and next process. | There is a slight effect on the component characteristics (spec irrelevant). |
| 1 | | | It is difficult to detect a failure, though there is some reluctance. | There is no effect. |

### 4.2.2. RPN Evaluation—O Criteria

Table 3 shows the O criteria, one of the RPN evaluation factors for fuel cell-based hybrid power systems. To precisely evaluate O, the evaluation criteria were classified into 1 (failure occurrence frequency), 2 (possibility of occurrence), 3 (high occurrence rate), and 4 (Cpk value). In the third stage, the high incidence rate was evaluated by applying the PPM(Parts Per Million) index and the Cpk statistical tool was used, which measures the ability of the process to produce output within the required specifications. Cpk represents the capability of the process. If both sides have specifications (upper and lower limits) and the center of the distribution does not match the median of both specifications, bias occurs. In order to prepare and evaluate the incidence criteria of the entire system in detail, evaluation criteria were divided into three stages and four stages. In general, the O is considered good when Cpk is 1.33 or greater for a system or a process. The method for obtaining Cpk is as follows [40].

**Table 3.** RPN criteria for occurrence.

| Class | Occurrence | | | |
| --- | --- | --- | --- | --- |
| | Evaluation Criteria 1 | Evaluation Criteria 2 | Evaluation Criteria 3 | Evaluation Criteria 4 |
| | Failure Occurrence Frequency | Possibility of Occurrence | High Occurrence Rate | Cpk Value |
| 10 | Very High relationship | Guaranteed occurrence | 1/2 = 500,000 PPM | Less than 0.33 |
| 9 | High relationship | | 1/3 = 333,000 PPM | 0.33↑ |
| 8 | Somewhat High relationship | Frequent occurrence | 1/8 = 125,000 PPM | 0.51↑ |
| 7 | Lower than high relationship | | 1/20 = 50,000 PPM | 0.67↑ |
| 6 | Higher than normal relationship | Occasional occurrence | 1/80 = 12,500 PPM | 0.83↑ |
| 5 | Normal relationship | | 1/400 = 2500 PPM | 1.00↑ |
| 4 | Lower than normal relationship | | 1/2000 = 500 PPM | 1.17↑ |
| 3 | Low relationship | Relatively infrequent occurrence | 1/15,000 = 66.67 PPM | 1.33↑ |
| 2 | Very low relationship | | 1/150,000 = 6.67 PPM | 1.50↑ |
| 1 | Almost no relationship | Almost no occurrence | 1 or less/1,500,000 = 0.66 PPM or less | 1.67↑ |

To get the value of Cpk, the capability index Cp is required. Cp is calculated to assess the degree of process capability. Cp can be obtained as Equation (1).

$$\text{Cp} = \frac{USL - LSL}{6\sigma} = \frac{Size\ of\ stanard}{Actual\ Process\ Scatter\ index}. \tag{1}$$

Here, USL: upper specification limit and LSL: lower specification limit.

The value of Cpk can be calculated from the measured data. If there is only an upper limit of the specification, if there is only a lower limit of the specification, it can be divided into a case where both the upper and lower limits of the specification, the calculation formula is as follows (2)–(4).

$$\text{Only upper limit of specification : } \text{Cpk} = \frac{USL - \overline{X}}{3\sigma}, \tag{2}$$

$$\text{Only lower limit of specification : } \text{Cpk} = \frac{\overline{X} - LSL}{3\sigma}, \tag{3}$$

$$\text{When both upper and lower limit are specified}: \text{Cpk} = (1-k) \times \text{Cp}, \tag{4}$$

where Cp is the capability index and K is the bias.

K is obtained as follows (5).

$$K = \frac{\frac{(USL+LSL)}{2} - \overline{X}}{\frac{(USL-LSL)}{2}}. \tag{5}$$

### 4.2.3. RPN Evaluation—D Criteria

Table 4 shows the D criteria, one of the RPN evaluation factors for fuel cell-based hybrid power systems. The evaluation criteria were divided into 1 (detectability), 2 (detection difficulty), and 3 (detailed description) to minimize ambiguity and ensure evaluation accuracy.

**Table 4.** RPN criteria for detection.

| Class | Detection | | |
|---|---|---|---|
| | **Evaluation Criteria 1** | **Evaluation Criteria 2** | **Evaluation Criteria 3** |
| | **Detectability** | **Detection Difficulty** | **Detailed Description** |
| 10 | Failure (problem) condition completely undetectable. | Not detectable by known methods. | No control measures able to detect failure type. |
| 9 | Failure (problem) condition undetectable. | Detection through indirect, uncertain, or unverified methods. | Very low detectability according to current system management. |
| 8 | In sensory evaluation, while macrography is possible, failure (problem) condition detection is difficult. | Detected in customer reliability test. | Low detectability according to system-wide management. |
| 7 | | Detected in internal reliability test. | Very low likelihood of detection. |
| 6 | Failure (problem) condition normally detected. | Detected in self-mount test. | Low likelihood of detection |
| 5 | | Detected in mass production test. | Less than 50% probability of detection. |
| 4 | Failure (problem) condition sufficiently detected. | Detected in component evaluation. | Detection probability slightly higher than normal, 50% or more. |
| 3 | | Detected in initial sample step. | Slightly high detectability. |
| 2 | Almost certainly automatically detected during the process. | Detected in design simulation. | Very high detectability. |
| 1 | | Detected in concept design. | Certainly detected. |

### 4.3. Evaluation Method for RPN Evaluation Items Using Kendall's Concordance Coefficient

First, the research hypothesis was established for the RPN evaluation items S, O, and D, and the evaluation items reestablished within the FMEA team were evaluated. Based on the results of the internal evaluation, the significance probability was compared to confirm the validity of the research hypothesis for the evaluation items. The process returned to the potential effect evaluation step if the research hypothesis was rejected. Here, 'P' indicates the significance probability, i.e., the probability that the null hypothesis occurs. The probability that the research hypothesis occurs is set to '1-P'; if the significance probability is less than 5%, then the null hypothesis is rejected, and the research hypothesis is supported. Table 5 shows the null and research hypotheses of this study [35].

**Table 5.** The null and research hypotheses of this study.

| |
|---|
| Research hypothesis: The evaluation scores by item of the evaluators will be similar, thus resulting in high reliability. |
| $H_0$: The evaluation scores by item of the evaluators will not be similar, thus resulting in low reliability. $H_1$: The evaluation scores by item of the evaluators will be similar, thus resulting in high reliability. |

$H_0$ is the null hypothesis, which refers to the already established hypothesis. $H_1$ is the research hypothesis, which negates the null hypothesis; it refers to the method of validating the established research hypothesis.

There are many ways to find correlations, but the most common correlation coefficients are Pearson, Kendall, and Spearman. For the FMEA evaluation items, a non-parametric test was applied instead of a parametric test because an analysis method that directly calculates the probability and statistically tests the data is appropriate regardless of the shape of the population. Pearson is basically used for the correlation analysis, but since it is a parametric test that shows correlations when variables are continuous data, one of the Kendall and Spearman's methods was used to apply nonparametric tests without linear correlation. Spearman generally has higher values than Kendall's correlation coefficient, but is sensitive to deviations and errors in the data. Therefore, Kendall's correlation coefficient was applied in this study because the sample size was small and the data dynamics were large.

The internal evaluation of the FMEA team confirmed the validity of the research hypothesis on the reestablished evaluation items, after which the Kendall's concordance coefficient was compared to determine the reliability of the evaluation items for the individual evaluations. Kendall's concordance coefficient indicates a correlation between multiple evaluators assessing the same sample. The coefficient ranges from 0 to 1, with a higher value indicating stronger correlations. Coefficients above 0.9 are generally considered to indicate very high concordance, meaning that the evaluators apply essentially the same criteria when evaluating the samples, decreasing the ambiguity of the evaluation items, removing evaluation arbitrariness, and encouraging objectivity [36]. If the coefficients for each item deviate from the criteria, the process returns to the potential effect evaluation step.

This study calculated Kendall's concordance coefficient using Equations (6)–(8) and Statistical Package for the Social Sciences (SPSS), a widely used program in statistical analysis. The coefficient was calculated to analyze the concordance between the evaluators for the reestablished S, O, and D evaluation results.

$$W = \left[ \frac{12 \sum T_i^2}{K^2 N(N^2 - 1)} \right] - \frac{3(N + 1)}{N - 1}, \tag{6}$$

where $T_I$ is the sum of the classes assigned to each target item by the evaluators, $K$ is the number of evaluators, and $N$ is the number of target items.

The formula for calculating $T_i$ finds the mean ($R_i$) for the sum of sequence scales.

$$\overline{R_i} = \frac{\sum R_i}{N}. \tag{7}$$

Then, the average deviation $T_i$ for each item can be obtained as follows.

$$T_i = \sum \left( R_i - \overline{R_i} \right)^2. \tag{8}$$

When establishing the evaluation items, the reliability of the internal evaluation results is verified using the significance probability for the research hypothesis for S, O, and D. The Kendall concordance coefficient was applied to reestablish the evaluation items that satisfy the criteria.

Based on the confirmed evaluation items, the external evaluators were requested to simultaneously evaluate both the existing and reestablished evaluation items. The significance probability and Kendall's concordance coefficient could again be applied to the results of the existing and reestablished evaluation

items to judge the application of the same standard. Thus, using the reestablished evaluation items, it is possible to verify that the evaluators are making objective, rather than arbitrary, decisions.

## 5. System Configuration and Subsystem Classification of the Hybrid Power System for Ships Subject to FMEA Evaluation

The fuel cell-based hybrid power system for ships consists of a power generation, power distribution, output performance verification, and control and management system. Figure 5 shows the subsystems of each system. The power produced by the power generation system is dispersed in the power distribution system, and the product of the output verification system is regulated according to the control and management system commands. The power distribution, output performance verification, and control and management systems have already been applied to all the ships currently under construction; hence, their operation reliability is sufficiently secured, and they were excluded from the FMEA of this study. Out of the subsystems of the hybrid power system, the power generation system (i.e., the failure mode and failure effect of the power source) was evaluated.

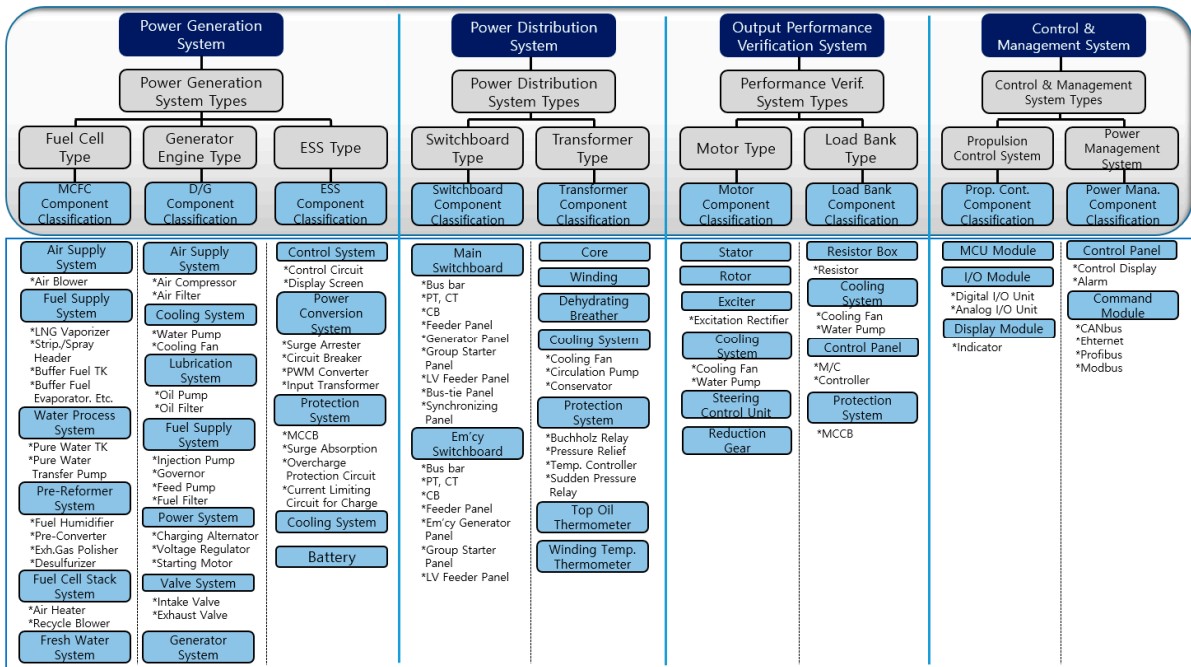

**Figure 5.** System configuration and subsystem classification of the hybrid power source for the ship.

### 5.1. Overall Composition of Power Generation System of the Hybrid Power System for Ships

The power generation system can be divided according to the power use purpose, as shown in Figure 6: main power, emergency power, auxiliary power, and alternative maritime power (AMP). The fuel cell, generator, and battery supply power to the main and auxiliary power sources. While AMP is generally supplied from onshore sources through cold ironing, in the hybrid power system, depending on the anchoring period, fuel cells with low greenhouse gas emissions can supply power on board.

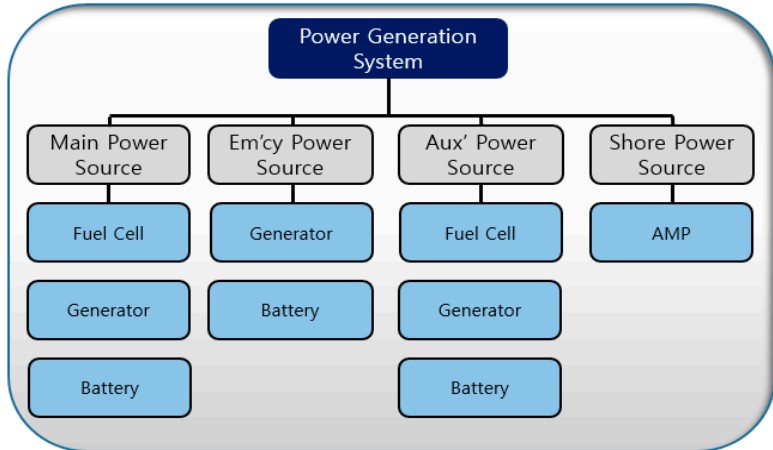

**Figure 6.** Power generation system of the hybrid power system for a ship.

## 5.2. Classification of Fuel Cell Components

A molten-carbonate fuel cell (MCFC) generally consists of a regulator, desulfurizer, humidifier, pre-converter, super heater, recycle blower, fresh air blower, inline heater, and catalytic oxidizer [41]. However, MCFCs for ships are comprised of the following components as shown in the block diagram of Figure 7: an air supply system, fuel supply system, water process system, pre-reformer system, fuel cell stack, fresh water system, auxiliary boiler and steam system, and cargo handling system [42].

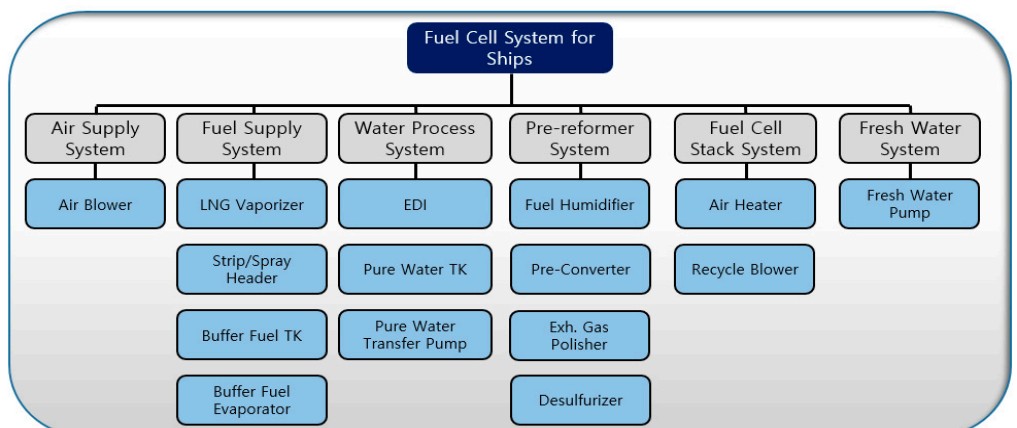

**Figure 7.** Fuel cell system of the hybrid power system for a ship.

Figure 8 is the configuration of the fuel cell for ships. The MCFC fuel supply system for ships must be connected to the pre-reformer system in the liquified natural gas (LNG) fuel supply chain, the fuel supply system of LNG propulsion ships was selected.

The electrolyte of the molten carbonate fuel cell (MCFC) is alkali metal carbonate, which is a mixture of lithium and potassium or lithium and sodium carbonate contained in a ceramic matrix of $LiAlO_2$. In general, it operates at a high temperature of 600–700 °C and carbonate ions ($CO_3{}^{2-}$) act as a charge carrier. Figure 9 and Equations (9)–(11) show a schematic diagram and chemical reactions occurring in MCFC [41].

$$\text{Total Reaction}: \ H_2 + \frac{1}{2}O_2 + CO_2 \rightarrow H_2O + CO_2. \tag{9}$$

$$\text{Anode Reaction}: \ H_2 + CO_3^{2-} \rightarrow CO_2 + H_2O + 2e^-. \tag{10}$$

$$\text{Cathode Reaction}: \ \frac{1}{2}o_2 + CO_2 + 2e^- \rightarrow CO_3^{2-}. \tag{11}$$

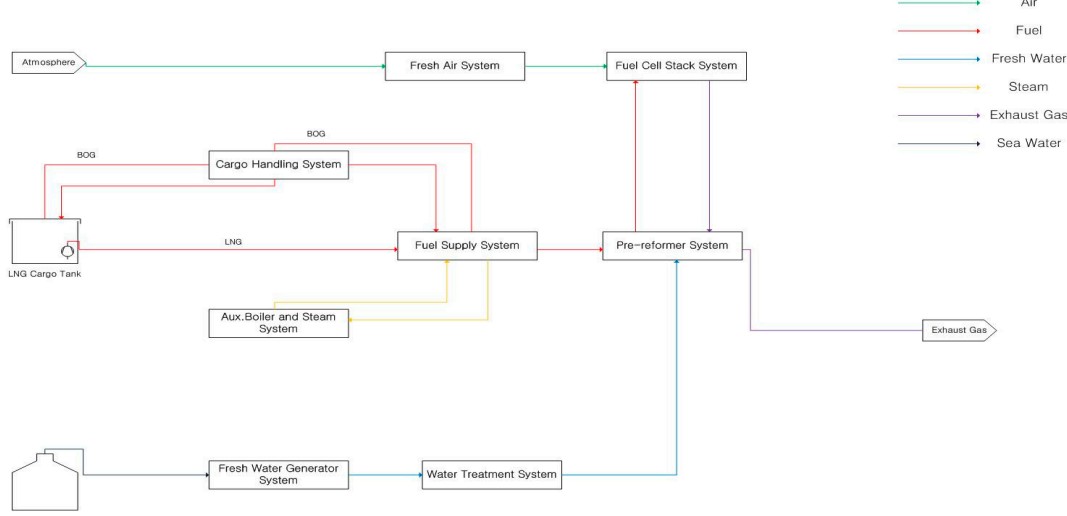

**Figure 8.** Composition of the fuel cell system for a ship.

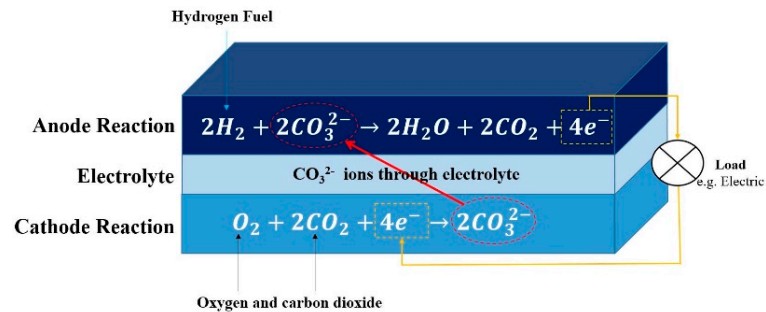

**Figure 9.** The schematic diagram and chemical reactions for the molten-carbonate fuel cell (MCFC) using hydrogen fuel.

MCFC needs to be supplied carbon dioxide together with oxygen to the cathode. The supplied carbon dioxide is converted into carbonate ions and becomes a means of moving ions between the cathode and the anode. The transferred carbonate ions are converted back to carbon dioxide by reaction with hydrogen at the anode side, and water and electricity are generated together as a result. In MCFC, not only hydrogen but also carbon monoxide can be used as fuel. Figure 10 schematic diagram and chemical reactions for MCFC using carbon monoxide fuel.

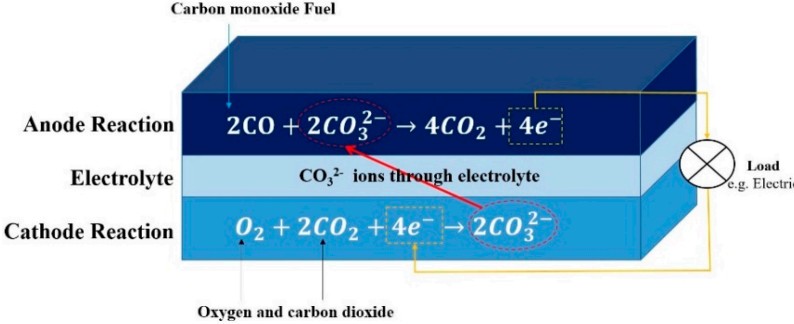

**Figure 10.** The schematic diagram and chemical reactions for the MCFC using carbon monoxide fuel.

In case of using carbon monoxide as fuel, the chemical reaction of the cathode is the same as that of using hydrogen as fuel. Oxygen and carbon dioxide supplied to the cathode react with each

other to be converted to carbonated ions, which are transferred to the anode through the electrolyte. The transferred carbonate reacts with carbon monoxide supplied to the anode side and is converted back to carbon dioxide.

### 5.3. Classification of Generator Engine Components

The systems of diesel engines, which have been most commonly used as driving generators, include the air supply, cooling, lubrication, fuel supply, power, and valve systems, while electric equipment includes electric governors, measuring equipment, control and safety devices, and cooling devices. In the case of governors in particular, electronic equipment is currently used. It receives the signal from the power management system (PMS) and adjusts the engine speed using torque control. Figure 11 shows the generator engine components.

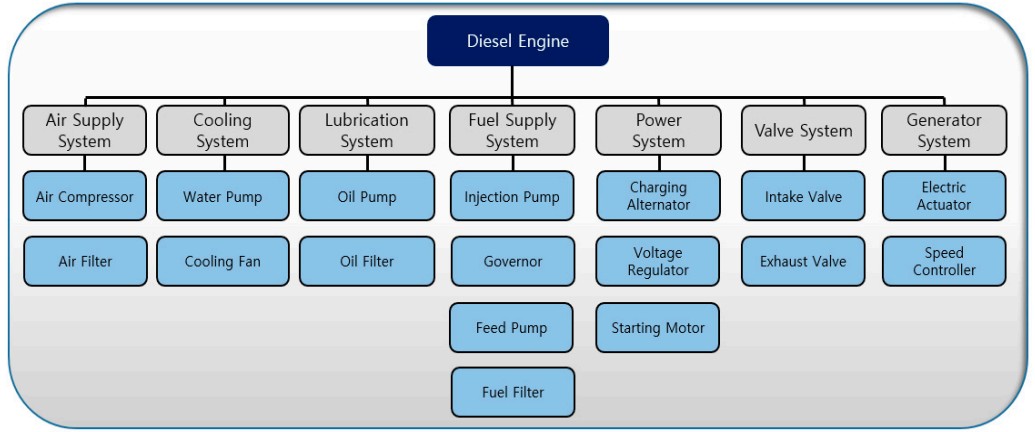

**Figure 11.** Diesel engine system of the hybrid power system for a ship.

### 5.4. Classification of Energy Storage System (ESS) Components

An energy storage system (ESS) is divided into control, cooling, protection, and power control systems. The protection system includes a reverse current protection device in the event of power failure, Molded Case Circuit Breaker (MCCB) and surge absorbing element, overcharge protection circuit, and charging current limiting circuit. The representative power conversion systems include pulse width modulation (PWM) converters and input transformers. Figure 12 shows the ESS components.

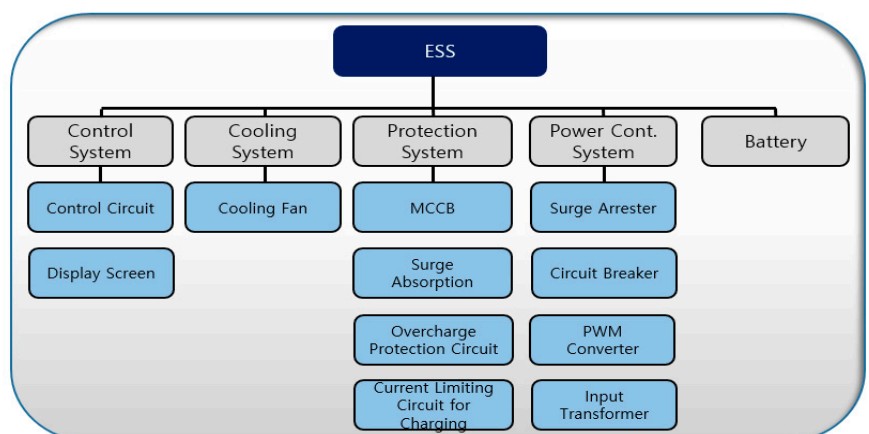

**Figure 12.** Energy storage system (ESS) of the hybrid power system for a ship.

## 6. Analysis of FMEA Performance Results

Before applying the fuel cell-based hybrid power system to actual ships, this study first performed FMEA to evaluate the system stability and reliability using onshore test beds. The types of failures that may occur in ship applications were identified, their effects were assessed, and corresponding improvements and supplements to the system were proposed.

The power produced from the hybrid power generation system was distributed through the power distribution system, passed through a synchronization system, and was converted to a voltage and frequency suitable for the output performance verification system. Table 6 shows the selected types of equipment required for a FMEA of the hybrid power system.

**Table 6.** Equipment list for FMEA of hybrid power system for ship.

| Upper System | Group | Subsystem | Subgroup | Equipment |
|---|---|---|---|---|
| Fuel cellSystem | 1 | Air supply system | 1.1 | Air blower |
| | | Water treatment system | 1.2 | Water pump, Water tank |
| | | Pre-reformer system | 1.3 | Desulfurizer |
| Diesel generator system | 2 | Air supply system | 2.1 | Air compressor, Air filter |
| | | Cooling system | 2.2 | Water pump |
| | | Lubrication system | 2.3 | Oil pump, Oil filter |
| | | Fuel supply system | 2.4 | Injection pump, Feed pump, Fuel filter |
| | | Power system | 2.5 | Charging alternator, Starting motor |
| | | Valve system | 2.6 | Intake valve |
| ESS system | 3 | Control system | 3.1 | Control circuit |
| | | Power conversion system | 3.2 | Surge arrester |
| | | Protection system | 3.3 | MCCB |
| | | Cooling system | 3.4 | Cooling fan |

### 6.1. FMEA Analysis Results of Fuel Cell System

Based on the FMEA results for the fuel cell system, three systems were examined from highest to lowest RPN, the results of which can be found below.

(1) Coating loss occurs due to the rapid ON/OFF desulfurizer cycle, and the desorption amount is reduced. Stack life is improved by replacing the adsorbent.
(2) Coating loss occurs due to the rapid ON/OFF desulfurizer cycle, blocking the back end desulfurizer filter. Stack life is improved by replacing the adsorbent.
(3) Initial power generation of the fuel cell is impossible due to the excessive flow of the air blower. A low air stoic supply is designed for the ignition of the oxidizer.

Even if the sulfur component contained in the natural gas is 0.2 ppm or less, the desulfurization process is required because the activity of the steam reforming catalyst is lowered and the electrode in the MCFC is poisoned, thereby greatly reducing the performance. Desulfurization methods include hydrogen desulfurization (HDS) and the use of absorbents for desulfurization. The method mentioned in this paper is the use of absorbents, which use activated carbon to absorb and remove sulfur. It is coated with a catalyst to enhance the absorption of sulfur. If this coating is not sufficient, the performance of the absorbent may be degraded. Table 7 is the FMEA results of the ship MCFC system [41].

**Table 7.** The FMEA results of the ship MCFC system.

| No | Ref. No. | Component | Failure Mode | Failure Effect | Cause of Failure | Severity | Occurrence | Detection | RPN | Recommended Measures |
|----|----------|-----------|--------------|----------------|------------------|----------|------------|-----------|-----|----------------------|
| | | | | | | | | | | |
| | Item Information | | | | | | | | | |
| | | | | MCFC System | | | | | | |
| 1 | 1.1 | Air blower | Insufficient air flow | Oxidizer temperature limit | Insufficient air flow | 8 | 2 | 4 | 64 | The air flow required for oxidizer operation is calculated, and the pump is selected. |
| | | | Insufficient air flow | Air stoic, Resonance generated due to inconsistency | Insufficient air flow | 6 | 6 | 3 | 108 | A larger supply is designed than the theoretical air flow required for oxidizer operation. |
| | | | Insufficient air flow | Incomplete combustion | Insufficient air flow | 9 | 5 | 3 | 135 | A larger supply is designed than the theoretical air flow required for oxidizer operation. |
| | | | Excessive air flow | Initial ignition impossible | Excessive air flow | 8 | 5 | 4 | 160 | A smaller air stoic supply is designed for the initial ignition of the oxidizer. |
| | | | Insufficient air flow | Increased CO concentration in reformate | Insufficient air flow | 8 | 6 | 3 | 144 | The air flow relative to the amount of reformate that the prox must process is calculated, and the pump is selected. |
| | | | Excessive air flow | Pump power consumption increases | Excessive air flow | 4 | 3 | 3 | 36 | The air flow relative to the amount of reformate that the prox must process is calculated, and the pump is selected. |
| | | | Excessive air flow | Prox reaction performance decreases due to fast flow rate, increasing CO | Excessive air flow | 8 | 3 | 3 | 72 | The air flow relative to the amount of reformate that the prox must process is calculated, and the pump is selected. |
| | | | Insufficient air flow | Increased catalytic CO poisoning | Insufficient air flow | 8 | 3 | 3 | 72 | Select flow rate using the calculated value compared to the amount of hydrogen in the reformate supplied to the stack. |
| | | | Excessive air flow | Catalytic oxidation reduces the anode performance and lifetime | Excessive air flow | 8 | 3 | 3 | 72 | Select flow rate using the calculated value compared to the amount of hydrogen in the reformate supplied to the stack. |
| | | | Insufficient air flow | Stack output and lifetime reduction | Insufficient air flow | 8 | 5 | 4 | 160 | Select flow rate using the calculated value compared to the amount of hydrogen in the reformate supplied to the stack. |
| | | | Excessive air flow | Pump power consumption increases | Excessive air flow | 4 | 3 | 4 | 48 | Select flow rate using the calculated value compared to the amount of hydrogen in the reformate supplied to the stack. |

**Table 7.** *Cont.*

| | Item Information | | Failure Mode | Failure Effect | Cause of Failure | Severity | Occurrence | Detection | RPN | Recommended Measures |
|---|---|---|---|---|---|---|---|---|---|---|
| No | Ref. No. | Component | | | | | | | | |
| | | | | | MCFC System | | | | | |
| | 1.2 | Water pump | Insufficient water flow | Reformer output reduction | Insufficient water flow | 6 | 6 | 4 | 144 | Calculate and supply water amount required for reforming. |
| | | | Insufficient water flow | Coking generated in reformer | Insufficient water flow | 7 | 6 | 4 | 168 | Calculate and supply water amount required for reforming. |
| | | | Excessive water flow | Degradation of reformer performance due to reduced reformer temperature | Excessive water flow | 5 | 5 | 4 | 100 | Calculate and supply water amount required for reforming. |
| | 1.3 | Desulfurizer | Coating loss | Decrease in sulfur adsorption | Rapid ON/OFF cycle | 6 | 5 | 7 | 210 | Adsorbent replacement. |
| | | | | Filter blocked in back end of desulfurizer | Rapid ON/OFF cycle | 6 | 5 | 7 | 210 | Adsorbent replacement. |
| | | | Catalyst crack | Reactor pressure drop increase | Rapid ON/OFF cycle and mechanical shock | 8 | 3 | 5 | 120 | Adsorbent replacement. |
| | | | | Decrease in sulfur adsorption | Rapid ON/OFF cycle and mechanical shock | 8 | 3 | 5 | 120 | Adsorbent replacement. |
| | | | | Filter blocked in back end of reactor | Rapid ON/OFF cycle and mechanical shock | 8 | 3 | 5 | 120 | Adsorbent replacement. |
| | | | Olefin adsorption | Increase of S concentration in fuel | Excess olefin concentration in fuel | 5 | 3 | 5 | 75 | Filter replacement. |
| | | | | Deterioration of filter life | Excess olefin concentration in fuel | 5 | 3 | 5 | 75 | Filter replacement. |
| | | | Water adsorption | Increase of S concentration in fuel | Pre-filter performance deterioration | 7 | 3 | 5 | 105 | Filter replacement. |
| | | | | Deterioration of filter life | Pre-filter performance deterioration | 7 | 3 | 5 | 105 | Filter replacement. |
| | | | Catalyst crack | Reactor pressure drop increases | Sudden change in fuel pressure | 8 | 3 | 5 | 120 | Filter replacement. |
| | | | | Filter blocked in back end of reactor | Sudden change in fuel pressure | 8 | 3 | 5 | 120 | Filter replacement. |

*6.2. FMEA Analysis Results of Diesel Generator System*

Based on the FMEA results for the diesel generator system, three systems were examined from highest to lowest RPN, the results of which can be found below.

(1) If the engine power is insufficient due to the inability of the engine to remove impurities in the fuel filter, and the situation persists, engine wear and cracks occur. The fuel filter must be cleaned and replaced frequently to prevent this.
(2) The engine could not be started due to the failure of the starting switch, starting relay, or magnetic kick switch of the starting motor, leading to a dead ship state. To prevent this, the starting motor was disassembled and components were replaced periodically.
(3) Owing to the aging of the air filter, the air intake to the engine was insufficient, and the engine could not be started, leading to a dead ship state. To prevent this, the air filter was frequently cleaned and replaced.

Table 8 is the FMEA results of diesel generator system.

*6.3. FMEA Analysis Results of ESS System*

Based on the FMEA results for the ESS system, three systems were examined from highest to lowest RPN, the results of which can be found below.

(1) Insulation resistance functionality deteriorated due to soot and metal particles attaching to the MCCB, which might damage the electric equipment at the MCCB back end. In this situation, the MCCB was replaced immediately.
(2) Owing to the control failure of the cooling fan, the electrolyte temperature rose, and the battery capacity was reduced. The ambient temperature should be decreased, and the specific gravity of the electrolyte should be adjusted.
(3) Due to the adjustment failure of the cooling fan, the electrolyte temperature rose, and separator aging and internal short circuiting occurred. To prevent this, the separator should be replaced.

Table 9 is the FMEA results of ESS.

**Table 8.** The FMEA results of the ship diesel generator system.

| No | Ref. No. | Component | Failure Mode | Failure Effect | Cause of Failure | Severity | Occurrence | Detection | RPN | Recommended Measures |
|---|---|---|---|---|---|---|---|---|---|---|
| | | | | | **Diesel Generator System** | | | | | |
| 2 | 2.1 | Air compressor | Unable to start engine | Dead ship | Insufficient compression pressure | 6 | 2 | 3 | 36 | Repair and replace air compressor. |
| | | Air filter | Insufficient engine power | Air supply pump overload | Insufficient air intake | 5 | 4 | 4 | 80 | Clean or replace air filter. |
| | | | Unable to start engine | Dead ship | Insufficient air intake | 8 | 3 | 5 | 120 | Clean or replace air filter. |
| | 2.2 | Water pump | Overheating | Engine wear and tear, cracks | Insufficient coolant transfer | 7 | 2 | 5 | 70 | Inspect cooling valve or inspect or repair pump. replacement |
| | 2.3 | Oil filter | Engine knocking | Cylinder aging | Unable to remove impurities | 8 | 4 | 3 | 96 | Clean or replace oil filter. |
| | 2.4 | Injection pump | Insufficient engine power | Engine cracks | Insufficient fuel injection | 7 | 3 | 4 | 84 | Adjust injection pump. |
| | | | Abnormal idle operation | Injection nozzle cracks | Air intake in injection pump | 5 | 3 | 4 | 60 | Remove air in pump. |
| | | | Excessive fuel consumption | Knocking due to rich burn | Excessive fuel injection | 6 | 3 | 3 | 54 | Adjust injection pump. |
| | | Feed pump | Insufficient engine power | Knocking | Pump functionality deterioration | 6 | 3 | 5 | 90 | Repair or replace pump. |
| | | Fuel filter | Insufficient engine power | Engine wear and tear, cracks | Unable to remove impurities | 7 | 4 | 5 | 140 | Clean or replace fuel filter. |

**Table 8.** *Cont.*

| No | Item Information Ref. No. | Item Information Component | Failure Mode | Failure Effect | Cause of Failure | Severity | Occurrence | Detection | RPN | Recommended Measures |
|---|---|---|---|---|---|---|---|---|---|---|
| | | | | | **Diesel Generator System** | | | | | |
| | 2.5 | Charging alternator | Unable to start engine | Dead ship | Electrical wiring slack and short circuit | 8 | 2 | 3 | 48 | Retighten or replace charging alternator. |
| | | Starting Motor | Unable to start engine | Dead ship | Starting switch failure, starting relay failure, magnetic switch failure | 8 | 3 | 5 | 120 | Disassemble starting motor. |
| | 2.6 | Intake valve | Insufficient engine power | Engine wear and tear, cracks | Incorrect valve clearance | 5 | 5 | 3 | 75 | Adjust intake valve. |
| | | | | Engine wear and tear, cracks | Poor valve adhesion | 6 | 4 | 3 | 72 | Repair intake valve. |
| | | | Insufficient compression pressure | Trouble starting | Poor valve closure | 7 | 2 | 5 | 70 | Analyze fuel injection timing and replace valve. |
| | | | | | Valve spring damage | 7 | 4 | 3 | 84 | Replace valve spring. |

**Table 9.** The FMEA results of the ESS.

| No. | Ref. No. | Component | Failure Mode | Failure Effect | Cause of Failure | Severity | Occurrence | Detection | RPN | Recommended Measures |
|---|---|---|---|---|---|---|---|---|---|---|
| | | | | | ESS System | | | | | |
| 3 | 3.1 | Control circuit | Control failure | Electrolyte leakage and reduction | Overcharge | 8 | 2 | 5 | 80 | Replenish purified water and adjust electrolyte volume and specific gravity. |
| | | | Control failure | Pole plate corrosion | Overcharge | 8 | 2 | 4 | 64 | Control floating charge voltage or shorten equalization time. |
| | | | Control failure | Pole plate bending and active material drop | Overcharge | 8 | 2 | 5 | 80 | Inspect charging current and ambient temperature. |
| | 3.2 | Surge arrester | Electrical breakdown | PCS function loss | Internal short circuit | 7 | 2 | 5 | 70 | Replace surge arrester. |
| | 3.3 | MCCB | Insulation resistance deterioration | Damage to back end electric equipment | Overcurrent blocking soot, metal particle adhesion | 8 | 2 | 7 | 112 | Immediately replace MCCB. |
| | 3.4 | Cooling fan | Fan rpm adjustment failure | Internal short circuit due to separator aging | Electrolyte temperature increase | 7 | 3 | 5 | 105 | Replace separator. |
| | | | Fan rpm adjustment failure | Foaming at full charge | Electrolyte temperature increase | 7 | 2 | 4 | 56 | Control floating charge voltage or shorten equalization time. |
| | | | Fan rpm adjustment failure | Battery capacity reduction | Electrolyte temperature increase | 3 | 6 | 6 | 108 | Improve ambient temperature and adjust electrolyte specific gravity. |
| | | | Fan rpm adjustment failure | Pole plate bending and active material drop | Electrolyte temperature increase | 5 | 3 | 4 | 60 | Improve ambient temperature and adjust electrolyte specific gravity. |

### 6.4. FMEA Results for Each System

This study precisely identified the hybrid power system failure types and applied the reestablished RPN criteria to analyze the potential effects of failure. This study sought to derive consistent results between evaluators through newly applied evaluation criteria, obtaining results that could confirm safety and reliability when applied to the hybrid power system of a ship. Before applying Kendall's concordance coefficient, the hypothesis "The evaluation scores by item of the evaluators will be similar" was established according to the research objective. The significance probability between the existing and reestablished evaluation items was compared, confirming the validity of the research hypothesis. Kendall's concordance coefficient was applied using SPSS to confirm the concordance rate of the evaluation results between the existing and reestablished evaluation items as shown in Tables 10–12.

**Table 10.** Comparison of severity evaluation results between existing and reestablished evaluation items.

| Test Statistics of Existing Evaluation Items | | Test Statistics of Reestablished Evaluation Items | |
| --- | --- | --- | --- |
| K | 3 | K | 3 |
| N | 50 | N | 50 |
| Approximate significance probability | 0.000 | Approximate significance probability | 0.000 |
| Kendall's W | 0.700 | Kendall's W | 0.906 |

**Table 11.** Comparison of occurrence evaluation results between existing and reestablished evaluation items.

| Test Statistics of Existing Evaluation Items | | Test Statistics of Reestablished Evaluation Items | |
| --- | --- | --- | --- |
| K | 3 | K | 3 |
| N | 50 | N | 50 |
| Approximate significance probability | 0.000 | Approximate significance probability | 0.000 |
| Kendall's W | 0.703 | Kendall's W | 0.844 |

**Table 12.** Comparison of detection evaluation results between existing and reestablished evaluation items.

| Test Statistics of Existing Evaluation Items | | Test Statistics of Reestablished Evaluation Items | |
| --- | --- | --- | --- |
| K | 3 | K | 3 |
| N | 50 | N | 50 |
| Approximate significance probability | 0.002 | Approximate significance probability | 0.000 |
| Kendall's W | 0.565 | Kendall's W | 0.861 |

In this study, external evaluators assessed the same samples; based on the significance probability for the evaluation results of each item, the research hypothesis was supported. In addition, among the RPN items, the Kendall's concordance coefficient was 0.906 for S, 0.844 for O, and 0.861 for D. Compared to the existing evaluation items, the results for the reestablished evaluation items indicated that each evaluator applied essentially the same criteria when assessing the samples. The reliability of the evaluation results was therefore verified, and criteria for providing countermeasures for each failure mode were established based on the detected results.

To establish the criteria for countermeasures according to the RPN results of the fuel cell-based hybrid power system, it must be decided whether the absolute or relative RPN values will be used as the standard. To establish countermeasures via relative RPN values, the conditions of the targets for comparison must be similar (e.g., the number of items and the content of each item). However, as the internal device configurations and characteristics differ for each system, the number of evaluation items and the type and contents of each item also differ, making it difficult to apply relative criteria.

Therefore, this study defined the criteria of the reestablished evaluation items for countermeasures using absolute RPN values; specifically, the RPN evaluation class was defined as 1–10, and $1 \leq RPN \leq$

1000. The following were set as the criteria for establishing countermeasures assuming a reliability of 90%: RPN of 100 or more, and either S, O, or D was 8 or more. Table 13 shows the number of items that should be set for each system according to the criteria.

**Table 13.** The ration of items required to establish countermeasures for a system based on these criteria.

| System | Total Items | Number of Failure Modes to Establish Countermeasures | |
|---|---|---|---|
| MCFC | 25 | RPN 100 or more | 17 |
| | | 8 or more for each | 13 |
| Diesel Generator | 16 | RPN 100 or more | 3 |
| | | 8 or more for each | 4 |
| ESS | 9 | RPN 100 or more | 3 |
| | | 8 or more for each | 4 |
| Overall hybrid power system | 50 | - | - |

## 7. Conclusions

This study conducted a FMEA to evaluate the safety and reliability of a fuel cell-based hybrid power system for ships. Unlike diesel engines that are mainly used as propulsion power sources in conventional ships, new FMEA evaluation criteria and items are needed to apply fuel cell-based hybrid power sources to ships. In the RPN evaluation currently applied to shipbuilding in shipyards, existing RPN evaluations, the evaluation items and criteria are vaguely established; therefore, results for the same evaluation would differ vastly between evaluators. Accordingly, for the FMEA of this study, the evaluation was performed using several external evaluators who applied reestablished evaluation criteria that mitigate RPN evaluation problems. To analyze the concordance of the reestablished evaluation items, a research hypothesis was established, and the significance probabilities and Kendall's concordance coefficient were calculated using SPSS. The concordance coefficient was 0.906 for S, 0.844 for O, and 0.861 for D. The results indicate that each evaluator applied essentially the same criteria when evaluating the samples, demonstrating that the reliability of the evaluation results was high. The criteria used to establish countermeasures for each failure mode were set based on the D results of the evaluation.

Although having the same evaluation configuration for each hybrid power system is essential to establish countermeasures, each system contains different devices and characteristics, therefore, the number and type of evaluation items also differ. Since it is difficult to apply relative criteria, this study instead used absolute RPN values to set the criteria for establishing countermeasures: a RPN of 100 or more and an S, O, or D of 8 or more.

For the FMEA of this study the power generation system of the hybrid power system (i.e., the failure mode and failure effect of the power source) was evaluated. However, future research must conduct FMEA for the entire set of systems including the power generation, power distribution, output performance verification, and control and management systems of hybrid power systems. Future studies must also perform FMEA for different system operation modes (e.g., single and hybrid operation) to identify hazards that may arise in the systems of actual ships during operation. However, in spite of these limitations, the results of this study showed significant results as an evaluation to confirm the stability and reliability for applying a fuel–cell based hybrid power source to several ships.

**Author Contributions:** Conceptualization, J.K. and H.J.; Methodology, J.K., H.J. and K.P.; Formal analysis, H.J.; Software, J.K. and H.J.; Writing-original draft preparation, H.J.; Writing-review and editing, J.K., H.J. and K.P. All authors have read and agreed to the published version of the manuscript.

**Funding:** This research received no external funding.

**Conflicts of Interest:** The authors declare no conflict of interest.

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
