# Peer review of "Comparison and Verification of Reliability Assessment Techniques for Fuel Cell-Based Hybrid Power System for Ships"

_jmse, doi:10.3390/jmse8020074_

Round 1
Reviewer 1 Report
Abstract is a bit hard to follow. Especially, the work done and its motivation is not clearly mentioned. Some details such as the formed team are given, but more structured explanation is required. IMO has done this, GBS are introduced. But these need to be evaluated by experts. We did this and we tested the outputs of the workshop on fuel cells.
Although the title mentioned fuel cells, abstract mentions generators and other technologies. The reason is not clear.
Make sure acronyms are correct. FMEA, not FEMA. (page 1 line 15)
The references need to be distributed in a better way. Instead of [1-3] at the end of first paragraph, they need to be placed where they are relevant as [1], [2] and [3].
Referencing requires improvement. Instead of too long links, shortening services such as bit.ly can be used.
After reading the paper, two things are not clear, the work done in the paper and the contribution of the paper.
It is mentioned that a team of experts is utilized, but how many? how are they selected? why are they deemed experts and reliable? Such information is required to understand the sample size, its reliability and dependability. As it stands, it is only claimed some information is gathered from somewhere.
Contribution is a bigger concern. IT seems that authors only implemented well known FMEA techniques on a hybrid energy system for ships. While, such different implementation field may be interesting, there is not enough novelty or contribution to make it worthy of journal publication. What is novel about the paper? Why should it be published in a journal? These questions are not answered in the paper.
Currently, it looks like a paper with a very lenghty background explanation (the techniques etc.) and a very limited implementation/use case.
Author Response
Reviewers' comments:
Reviewer 1
We are very grateful for your line-by-line comments across the manuscript. Your comments are valuable and very helpful for revising and improving our paper, as well as the important guiding significance to our research. We have studied the comments carefully and have revised the manuscript accordingly, which we hope meet with approval.
We put our responses to your comments on the original manuscript as well as the main corrections in the paper and the responses to your official comments are as follows.
All changes were highlighted
1) Abstract is a bit hard to follow. Especially, the work done and its motivation is not clearly mentioned. Some details such as the formed team are given, but more structured explanation is required. IMO has done this, GBS are introduced. But these need to be evaluated by experts. We did this and we tested the outputs of the workshop on fuel cells. Although the title mentioned fuel cells, abstract mentions generators and other technologies. The reason is not clear.
Response:
Following your suggestion, we modified abstract. The reason why GBS was mentioned in the abstract, which was first created, is because the recent development of GBS based on the SLA at IMO, which can ensure the stability of the ship from the construction stage of the ship, is related to the intention of introducing FMEA. The main reason for implementing FMEA is to ensure stability and reliability from the design stage. However, as I said, I decided that this part is not important, so I modified the Abstract.
In addition, this paper proposes an FMEA method for evaluating the stability and reliability of an electric propulsion test bed using a hybrid power source based on fuel cells. In Europe, some coastal ships are operated by fuel cells, but they are not universal.
Therefore, the FMEA evaluation criteria or items are not set in detail at the shipyard, and even though the evaluation criteria or items are set, they are not specific. Therefore, in this paper, we set a certain standard and selected experts in each field. The selected expert group redefines the FMEA criteria and items for fuel cell hybrid electric propulsion testbeds for ship applications, and the results of the evaluation provided significant results.
“In order to secure the safe operation of the ship, it is crucial to closely examine the suitability from the design stage of the ship, and to set up a preliminary review and countermeasures for failures and defects that may occur during the construction process. In shipyards, the FMEA evaluation method using RPN is used in the shipbuilding process. In the case of the conventional RPN method, evaluation items and criteria are ambiguous, and subjective factors such as evaluator's experience and understanding of the system operate a lot on the same contents, resulting in differences in evaluation results. Therefore, this study aims to evaluate the safety and reliability for ship application of the reliability-enhanced fuel cell-based hybrid power system by applying the re-established FMEA technique. Experts formed an FMEA team to redefine reliable assessment criteria for the RPN assessment factors Severity (S), Occurrence (O), and Detection (D). Analyze potential failures of each function of MCFC system, battery system and diesel engine components of fuel cell-based hybrid power system set as evaluation targets to redefine the evaluation criteria, and the evaluation criteria were derived by identifying the effects of potential failures. In order to confirm the reliability of the derived criteria, the reliability of individual evaluation items was verified by using the significance probability used in statistics and the coincidence coefficient of Kendall. The evaluation was conducted to the external evaluators using the reestablished evaluation criteria. As a result of analyzing the correspondence according to the results of the evaluation items, the severity was 0.906, the incidence 0.844, and the detection degree 0.861. Improved agreement was obtained, which is a significant result to confirm the reliability of the reestablished evaluation results.”
2) Make sure acronyms are correct. FMEA, not FEMA. (page 1 line 15)
Response:
Thank you for your detailed feedback. By editing the content of abstract, the letter was deleted.
3) The references need to be distributed in a better way. Instead of [1-3] at the end of first paragraph, they need to be placed where they are relevant as [1], [2] and [3].
Response:
As per your comment, we changed the expression way of relevant number.
“~the underlying causes of marine accidents and environmental pollution from ship construction and to prioritize ship safety[1],[2] and[3].”
“However, FMEA is the most common way to evaluate device reliability[4],[5].”
“~a method based on fuzzy logic that considers the interdependence between various failure modes[10],[11],[12],[13] and[14],~” etc.
4) Referencing requires improvement. Instead of too long links, shortening services such as bit.ly can be used.
Response:
According to your opinion, The long links among the references mentioned on the Internet were shortened using bit.ly.
“Maritime Safety Committee(MSC) 101 session. Available online: http://bitly.kr/U7R2WN6 (accessed on 22 November 2019).”
“Trends in IMO for Ship Safety and Marine Environment Protection. Available online: http://bitly.kr/WgqktaA4 (accessed on 01 November 2019).”
“IMO International Maritime Policy Trend. Available online: http://bitly.kr/NuQogc3i (accessed on 18 November 2019).”
“Correlation(Pearson, Kendall, Spearman). Available online: http://bitly.kr/smNl299P (accessed on 01 December 2019).”
“Analysis techniques for system reliablility Procedure for failure mode and effects analysis(FMEA). Available online: http://bitly.kr/DlQDoqSw (accessed on 30 November 2019).”
“Failure Mode and Effect Analysis (FMEA) of Redundant Systems. Available online: http://bitly.kr/7eSXMyz (accessed on 03 December 2019).”
“Guidance on failure modes and effects analysis(FMEA). Available online: http://bitly.kr/TZKtc45x (accessed on 02 December 2019)”
5) After reading the paper, two things are not clear, the work done in the paper and the contribution of the paper.
Response:
This paper proposes an FMEA method for evaluating the stability and reliability for the application of an electric propulsion test bed using a hybrid power source based on fuel cells.
In order to apply environmentally friendly ships, ships with fuel cells are operating mainly on small coastal ships.
These vessels are very different from the diesel engines used as existing ship power sources, so new FEMA evaluation criteria and items must be in place to apply these vessels.
However, even in shipyards that are currently building ships, the FMEA evaluation criteria or items have not been set in detail.
Therefore, in this paper, experts in the relevant fields were selected to establish the reestablished evaluation criteria and items. The selected expert group redefines the FMEA criteria and items for fuel cell hybrid electric propulsion testbeds for ship applications, and the results of the evaluation provided significant results.
We believe that the evaluation criteria and items derived from this study will improve the stability and reliability of fuel cell-based hybrid power sources that will be built later.
6) It is mentioned that a team of experts is utilized, but how many? how are they selected? why are they deemed experts and reliable? Such information is required to understand the sample size, its reliability and dependability. As it stands, it is only claimed some information is gathered from somewhere.
Response:
In this study, the FMEA team was formed to determine the criteria and item for FMEA using the fuel cell-based hybrid ship power source with RPN technique.
The FMEA team consisted of a total of 10 people. The selected experts are currently working in shipyards, research institutes, classification societies, engine makers, test and certification institutes, educational institutions, etc. with more than five years experience in the fuel cell, battery, and diesel engine systems fields. And We selected candidates who have more than 10 times experiences in evaluating FMEA.
Since we have been working in the relevant field for a certain period of time and have formed an FMEA team composed of experts with basic experience in FMEA evaluation, we recognize the problems with existing FMEA and know that the evaluation criteria and item setting are important.
In order to confirm the reliability of the evaluation criteria and items derived from the FMEA team, the reliability was verified by applying the significance probability and Kendall's matching coefficient before moving on to the next step for each RPN element.
If you request a profile from an FMEA team experts, we’ll send it to you.
7) Contribution is a bigger concern. It seems that authors only implemented well known FMEA techniques on a hybrid energy system for ships. While, such different implementation field may be interesting, there is not enough novelty or contribution to make it worthy of journal publication. What is novel about the paper? Why should it be published in a journal? These questions are not answered in the paper.
Response:
Thank you for your opinion. As mentioned earlier, this paper proposes an FMEA method for evaluating the stability and reliability for ship application of an electric propulsion test bed with a hybrid power source based on fuel cells.
Since the diesel engine used as a ship power source and the fuel cell-based hybrid power source applied in this paper are different in concept, new FEMA evaluation criteria and items should be provided to evaluate the new system.
Therefore, in this paper, the selected experts are selected through the criteria of experts in the field in order to establish the reestablished evaluation criteria and items.
The FMEA team, composed of a group of experts, redefines the FMEA criteria and items for fuel cell hybrid electric propulsion testbeds for ship applications.
The results of the evaluation provided significant results for the study. We believe that the evaluation criteria and items derived from this study will improve the stability and reliability of fuel cell-based hybrid power sources that will be built later.
8) Currently, it looks like a paper with a very lenghty background explanation (the techniques etc.) and a very limited implementation/use case.
Response:
Thank you for your advice. There has not been much research on FMEA for ship application yet, and the hybrid electric propulsion system using fuel cells is expected to attract attention with the introduction of eco-friendly ships. In light of the comments, the next study will attempt to explore further areas of research.
Reviewer 2 Report
Dear Authors,
Thank you for submitting your work to Journal of Marine Science and Engineering. Overall quality of this paper is good, especially for the introduction part to FMEA and RPN, but it cannot be published with current form. Here are some suggestions:
1) The font and paragraph format should be carefully checked, and grammar errors should be eliminated as many as possible.
1a) Grammar Error from Line 69-71 should be fixed.
1b) Font font and paragraph format error in Line 97 should be fixed.
1c) Paragraph format error in Line 107,108 should be fixed.
1d) Inconsistent table format and font in Table 10-12 (compared to all other tables) should be avoided.
etc.
Please read through your paper thoroughly again.
2) In line 328, the "water management system" is strange. It does not appear in graph or other places in the article. I think it should be corrected to "water process system", as water management in the fuel cell stack is normally required for low temperature fuel cell, like PEMFC, due to the excess liquid water. Molten Carbonate Fuel Cell produces vapor water and does not have this issue.
3) Result parts in Section 6 should be expanded.
4) Since your article is "Fuel Cell-based", fundamentals of fuel cell and basic components of single fuel cell stack should be included in introduction, e.g. chemical reactions, anode, cathode, electrolyte, flow field, etc. Roles of these basic components should be stated briefly as well.
Recent publications should be included.
Some recent publication examples for flow field from MDPI:
a) Xuyang Zhang et al., Experimental Studies of Effect of Land Width in PEM Fuel Cells with Serpentine Flow Field and Carbon Cloth. Energies 2019, 12, 471
b) Xin Luo et al., Numerical Simulation of a New Flow Field Design with Rib Grooves for a Proton Exchange Membrane Fuel Cell with a Serpentine Flow Field. Appl. Sci. 2019, 9, 4863.
5) Which part of Desulfurizer got Coating loss? The details of coating loss should be stated more clearly, since it plays an important role in your article.
Author Response
Reviewers' comments:
Reviewer 2
We are very grateful for your line-by-line comments across the manuscript. Your comments are valuable and very helpful for revising and improving our paper, as well as the important guiding significance to our research. We have studied the comments carefully and have revised the manuscript accordingly, which we hope meet with approval.
We put our responses to your comments on the original manuscript as well as the main corrections in the paper and the responses to your official comments are as follows.
All changes were highlighted
Thank you for submitting your work to Journal of Marine Science and Engineering. Overall quality of this paper is good, especially for the introduction part to FMEA and RPN, but it cannot be published with current form. Here are some suggestions:
1) The font and paragraph format should be carefully checked, and grammar errors should be eliminated as many as possible.
Response:
Thank you for your comment. We checked grammar, font and format and modified it.
1a) Grammar Error from Line 69-71 should be fixed.
“This FMEA study ensures safety and reliability of ship application with applying a fuel cell-based hybrid power system to ships (MCFC[100kW], A battery[30kW] with diesel generator[50kW]) test bed system was constructed to demonstrate experiment on commercial ship.”
1b) Font font and paragraph format error in Line 97 should be fixed.
“Figure 1 illustrates the typical FMEA process.”
1c) Paragraph format error in Line 107,108 should be fixed.”
“Figure 2 shows the meaning of each RPN item. S affects the customer in relation to the process or product when a potential failure occurs.”
1d) Inconsistent table format and font in Table 10-12 (compared to all other tables) should be avoided.
etc.
Modified table format and font.
Please read through your paper thoroughly again.
Thank you for your detail comment.
2) In line 328, the "water management system" is strange. It does not appear in graph or other places in the article. I think it should be corrected to "water process system", as water management in the fuel cell stack is normally required for low temperature fuel cell, like PEMFC, due to the excess liquid water. Molten Carbonate Fuel Cell produces vapor water and does not have this issue.
Response:
Thank you for your detailed feedback, I modified the word from “water treatment system” to “water process system”.
As you mentioned, the water management system is a part of PEMFC system, a low-temperature fuel cell using polymer electrolyte, and the concept of water process system is more suitable word than water management system because MCFC is a high-temperature fuel cell.
“A molten-carbonate fuel cell (MCFC) generally consists of a regulator, desulfurizer, humidifier, pre-converter, super heater, recycle blower, fresh air blower, inline heater, and catalytic oxidizer. However, MCFCs for ships are comprised of the following components as shown in the block diagram of Figure 7: an air supply system, fuel supply system, water process system, pre-reformer system, fuel cell stack, fresh water system, auxiliary boiler and steam system, and cargo handling system.”
3) Result parts in Section 6 should be expanded.
Response:
Thank you for your comment, As you mentioned, I have added a description of the contribution of this paper to the conclusion.
“This study conducted a FMEA to evaluate the safety and reliability of a fuel cell-based hybrid power system for ships. Unlike diesel engines that are mainly used as propulsion power sources in conventional ships, new FMEA evaluation criteria and items are needed to apply fuel cell-based hybrid power sources to ships. In the RPN evaluation currently applied to shipbuilding in shipyards, existing RPN evaluations, the evaluation items and criteria are vaguely established; therefore, results for the same evaluation would differ vastly between evaluators.”~ ~ ~ “For the FMEA of this study the power generation system of the hybrid power system (i.e., the failure mode and failure effect of the power source) was evaluated. However, future research must conduct FMEA for the entire set of systems including the power generation, power distribution, output performance verification, and control and management systems of hybrid power systems. Future studies must also perform FMEA for different system operation modes (e.g., single and hybrid operation) to identify hazards that may arise in the systems of actual ships during operation. However, in spite of these limitations, the results of this study showed significant results as an evaluation to confirm the stability and reliability for applying a fuel-cell based hybrid power source to several ships.”
4) Since your article is "Fuel Cell-based", fundamentals of fuel cell and basic components of single fuel cell stack should be included in introduction, e.g. chemical reactions, anode, cathode, electrolyte, flow field, etc. Roles of these basic components should be stated briefly as well.
Response: We highly appreciate your valuable comment. As per your comments, We added the contents related to chemical reactions of MCFC to explain that fundamentals and basic components of single fuel cell stack as below.
“The electrolyte of Molten Carbonate Fuel Cell(MCFC) is alkali metal carbonate, which is mixture of lithium and potassium or lithium and sodium carbonate contained in a ceramic matrix of LiAlO2. In general, it operates at a high temperature of 600~700℃ and carbonate ions(CO32-) acts as a charge carrier. Figure 9 and Equation (2)~(4) shows schematic diagram and chemical reactions occurring in MCFC.
Figure 9. The schematic diagram and chemical reactions for MCFC using hydrogen fuel.
Total Reaction : (2)
Anode Reaction : (3)
Cathode Reaction : (4)
MCFC need to be supplied carbon dioxide together with oxygen to the cathode. The supplied carbon dioxide is converted into carbonate ions and becomes a means of moving ions between the cathode and the anode. The transferred carbonate ions are converted back to carbon dioxide by reaction with hydrogen at the anode side, and water and electricity are generated together as a result. In MCFC, not only hydrogen but also carbon monoxide can be used as fuel. Figure 10 schematic diagram and chemical reactions for MCFC using carbon monoxide fuel.
Figure 10. The schematic diagram and chemical reactions for MCFC using carbon monoxide fuel.
In case of using carbon monoxide as fuel, the chemical reaction of the cathode is the same as that of using hydrogen as fuel. Oxygen and carbon dioxide supplied to the cathode react with each other to be converted to carbonated ions, which are transferred to the anode through the electrolyte. The transferred carbonate reacts with carbon monoxide supplied to the anode side and is converted back to carbon dioxide.”
Recent publications should be included.
Response:
According to your opinion, we searched for related articles mentioned and added the contents.
Some recent publication examples for flow field from MDPI:
a) Xuyang Zhang et al., Experimental Studies of Effect of Land Width in PEM Fuel Cells with Serpentine Flow Field and Carbon Cloth. Energies2019, 12, 471 b) Xin Luo et al., Numerical Simulation of a New Flow Field Design with Rib Grooves for a Proton Exchange Membrane Fuel Cell with a Serpentine Flow Field. Sci.2019, 9, 4863.
5) Which part of Desulfurizer got Coating loss? The details of coating loss should be stated more clearly, since it plays an important role in your article.
According to your opinion, we inserted to articles mentioned.
“Even if the sulfur component contained in the natural gas is 0.2 ppm or less, the desulfurization process is required because the activity of the steam reforming catalyst is lowered and the electrode in the MCFC is poisoned, thereby greatly reducing the performance. Desulfurization methods include hydrogen desulfurization (HDS) and the use of absorbents for desulfurization. The method mentioned in this paper is the use of absorbents, which use activated carbon to absorb and remove sulfur. It is coated with a catalyst to enhance the absorption of sulfur. If this coating is not sufficient, the performance of the absorbent may be degraded. Table 7 is the FMEA results of the ship MCFC system.[40]”
Reviewer 3 Report
The paper has a very intesting subject in applied naval science. But the manuscript has some shortages, especially in methodology.
->In lines 189-199 the mathematical background is completely absent.
->Similarly in lines 247-249 the Cpk statistical tool should be described completely (especially the mathematical base). Additionally values of Cpk i.e. 1.33 should be justified.
-> In section 6.4 the difference between the existing evaluation items and reestablised one is not obvious (how have they been formed? what have the addiotional measures been taken? etc.)
-> In line 134 it has been written that "O is doubled, the RPN is not doubled", but this is not obvious.
Minnor errors:
-> In line 15 "FMEA" should be instead of "FEMA".
-> In table 5 the sentences are not completely readable (right side).
-> In line 301 "5." has been written twice.
Author Response
Reviewers' comments:
Reviewer 3
We are very grateful for your line-by-line comments across the manuscript. Your comments are valuable and very helpful for revising and improving our paper, as well as the important guiding significance to our research. We have studied the comments carefully and have revised the manuscript accordingly, which we hope meet with approval.
We put our responses to your comments on the original manuscript as well as the main corrections in the paper and the responses to your official comments are as follows.
All changes were highlighted
The paper has a very interesting subject in applied naval science. But the manuscript has some shortages, especially in methodology.
->In lines 189-199 the mathematical background is completely absent.
Response:
Thank you for your detail comment. According to your opinion, I mentioned the mathematical background of Kendall's introduction of the coincidence coefficient applied to verify the reliability of the reestablished evaluation criteria.
“In this paper, Kendall’s coefficient of consensus mentioned to verify the reliability of the evaluation items is one of the methods used in nonparametric statistics to analyze the relationship between phenomena measured on the sequence scale. Kendall’s coincidence coefficient is typically used for attribute agreement analysis, with coefficient values ranging from 0 to 1. Also, the higher the value of the coefficient, the stronger the association. If the coefficient is greater than 0.9, the relevance is considered very high and the high or significant Kendall’s coefficient means that the evaluators apply essentially the same standard when evaluation the sample.”
->Similarly in lines 247-249 the Cpk statistical tool should be described completely (especially the mathematical base). Additionally values of Cpk i.e. 1.33 should be justified.
Response:
Thank you for your detail comment. According to your opinion, I described the Cpk(Capability of Process, Katayori)
Cpk represents Capability of Process. If both sides have specifications(upper and lower limits) and the center of the distribution does not match the median of both specifications, bias occurs. Cpk represents the capability index taking into account possible biases in the actual distribution.
To get the value of Cpk, the capability index Cp is required. Cp is calculated to assess the degree of process capability. Cp can be obtained as:
Here, USL : Upper Specification Limit, LSL : Lower Specification Limit
Calculated Cp and next get the bias K. K can be found as B/A.
Finally, Cpk can be obtained by:
For example, let’s calculate. The conditions are as follows:
USL : 155, LSL : 145, CL(Middle point) :150, σ(Standard Deviation) : 1, K : 0.2
-> In section 6.4 the difference between the existing evaluation items and reestablised one is not obvious (how have they been formed? what have the addiotional measures been taken? etc.)
Response:
This paper proposes an FMEA method for evaluating the stability and reliability for the application of an electric propulsion test bed using a hybrid power source based on fuel cells.
Since the diesel engine used as a ship power source and the fuel cell-based hybrid power source applied in this paper are different in concept, new FEMA evaluation criteria and items should be provided to evaluate the new system.
However, as shown in figure 3(a), the problem with FMEA applying the conventional RPN method is that the evaluation results vary depending on the experience and understanding of the evaluator.
Therefore, in this paper, the selected criteria and items for the reestablished evaluation criteria were selected by experts in the relevant field, and the FMEA team composed of expert groups was used to describe the FMEA failure type for the fuel cell hybrid electric propulsion test bed for ship application. The assessment criteria and items have been re-established by pinpointing and analyzing the potential impact of failures.
In order to compare the evaluation results, we applied the probability of significance used in statistics and Kendall's coefficient of agreement.
Through the evaluation criteria and items derived from this study, we believe that the stability and reliability of the fuel cell based hybrid power source for next new building project will be improved.
-> In line 134 it has been written that "O is doubled, the RPN is not doubled", but this is not obvious.
Response:
This content is about the change of the calculated total RPN according to the change of individual RPN items, and tried to convey the fact that it is possible to evaluate individual evaluation factors, but it is difficult to evaluate the mutual effects. The information you mentioned was incorrectly filled out and deleted.
Minnor errors:
-> In line 15 "FMEA" should be instead of "FEMA".
Response:
We revised the contents mentioned while revising the Abstract contents.
-> In table 5 the sentences are not completely readable (right side).
Response:
As you said, the content is corrected so that you can see it.
Table 5. The null and research hypotheses of this study.
|
Research hypothesis: The evaluation scores by item of the evaluators will be similar, thus resulting in high reliability. |
|
: The evaluation scores by item of the evaluators will not be similar, thus resulting in low reliability. |
|
: The evaluation scores by item of the evaluators will be similar, thus resulting in high reliability. |
-> In line 301 "5." has been written twice.
Response:
As you mentioned, we have removed the duplicated content.
Round 2
Reviewer 1 Report
The authors only provided a response file but the responses are not incorporated into the manuscript. For instance, the novelty of the work or how the experts are selected. Responses to these questions need to be added to the text. Reviewer asks these questions, because they need to be answered in the paper.
Also, references are not fixed as requested. Changing 1-3 to 1 , 2, 3 is not the request. When several references are cited, these should be distributed in a paragraph so that relevant reference appears where a claim is made.
E.g.
Claim 1, Claim 2, claim 3 [1-3] is NOT acceptable.
Claim 1 [1], claim 2 [2], and some more text and claim 3 [3]. is the normal way of citation.
Author Response
Reviewers' comments:
Reviewer 1
We are very grateful for your line-by-line comments across the manuscript. Your comments are valuable and very helpful for revising and improving our paper, as well as the important guiding significance to our research. We have studied the comments carefully and have revised the manuscript accordingly, which we hope meet with approval.
We put our responses to your comments on the original manuscript as well as the main corrections in the paper and the responses to your official comments are as follows.
All changes were highlighted
1) The authors only provided a response file but the responses are not incorporated into the manuscript. For instance, the novelty of the work or how the experts are selected. Responses to these questions need to be added to the text. Reviewer asks these questions, because they need to be answered in the paper.
RESPONSE
Thank you for your detailed feedback. As per your comments in the introduction section of the paper, we have reinforced the novelty of previous studies.
“In particular, in recent years, in order to apply environmentally friendly ships, ships using hybrid fuel cells, batteries etc. are being operated mainly on small coastal ships. These vessel systems are very different from the diesel engines used as conventional ship power sources, so new FMEA evaluation criteria and items should be provided to evaluate the safety and reliability of such vessels. However, even in shipyards that are currently building vessels, FMEA evaluation criteria or items have not been specifically set.
Therefore, in this study, the proposed FMEA was conducted to secure the safety and reliability for applying the fuel cell-based(MCFC[100kW], battery[30kW] and diesel generator[50kW]) test bed to the actual ship. We analyzed various problems in evaluating RPN, which in mainly used in FMEA, and formed an FMEA expert team to select evaluation criteria and items. As a result, we developed a worksheet applying the reestablished RPN evaluation criteria, and applied Kendall’s coefficient of correspondence to the existing evaluation results and the reestablished evaluation results for objective determination of the reestablished evaluation criteria. It was confirmed that the reestablished assessment in the FMEA evaluation of the combined power source showed more reliable results. In addition, the criteria for establishing countermeasures based on the results of the FMEA were prepared, and the proposed evaluation method was found to be effective for the application of the assessment of the safety and reliability of the combined power source.”
We highly appreciate your valuable comment. As per your comments in the section 3 of the paper, we have reinforced the novelty of previous studies.
“The FMEA team is aware of the problems with existing FMEA because it has been working in the field for a certain period of time and selected experts with basic experience in FMEA evaluation. Therefore, we understand the importance of FMEA evaluation criteria and item setting.
To composition of the FMEA team and the criteria for selecting experts are as follows.
The FMEA team consists of 10 expertise for group The selected experts are currently employed in shipyards, research institutes, classificationsociety, engine makers, test and certification institutes, and educational institutions Over 5 years of experience in fuel cell, battery and diesel engine system Have more than 10 times experiences in evaluation FMEA”
2) Also, references are not fixed as requested. Changing 1-3 to 1 , 2, 3 is not the request. When several references are cited, these should be distributed in a paragraph so that relevant reference appears where a claim is made.
E.g.
Claim 1, Claim 2, claim 3 [1-3] is NOT acceptable.
Claim 1 [1], claim 2 [2], and some more text and claim 3 [3]. is the normal way of citation.
RESPONSE
Following your suggestion, we modified reference of the paper as below sentence.
“In the early 2000s, the Maritime Safety Committee (MSC) of the International Maritime Organization (IMO) adopted the item goal-based new ship construction standards (GBS)[1], which present new ship design and construction concepts for the long-term organizational work plan. They then developed safety level approach (SLA)-based GBS that are applicable to all ships[2]. The IMO has since actively strengthened the Safety of Life at Sea (SOLAS) standards based on the GBS to reduce the underlying causes of marine accidents and environmental pollution from ship construction and to prioritize ship safety[3].”
“FMEA, a type of risk assessment method, was developed for the Apollo project by the National Aeronautics and Space Administration (NASA) in the mid-1960s. Since then, three major US automakers have introduced their own assessment system “QS-9000”[4]. However, FMEA is the most common way to evaluate device reliability[5].”
“FMEA was first used in the NASA Apollo project in the 1960s. In 1974, it was used to develop United States Navy missiles and was established as the United States MIL-STD-1629 standard. Afterwards, the QS-9000 standard was established by the United States automobile industry, and FMEA was introduced in all industries, including shipbuilding[4]. The FMEA method prioritizes resources, ranks risks, and creates an activity and control plan to analyze the target system[5][6], thereby analyzing failure types and their influence and examining improvement measures with consideration of criticality[17].”
“In this paper, Kendall’s coefficient of consensus mentioned to verify the reliability of the evaluation items is one of the methods used in nonparametric statistics to analyze the relationship between phenomena measured on the sequence scale[35]. Kendall’s coincidence coefficient is typically used for attribute agreement analysis, with coefficient values ranging from 0 to 1. Also, the higher the value of the coefficient, the stronger the association. If the coefficient is greater than 0.9, the relevance is considered very high and the high or significant Kendall’s coefficient means that the evaluators apply essentially the same standard when evaluation the sample[36].”
“In the FMEA performance step, the causes, effects, countermeasures, and severity for each failure mode are discussed; these items are recorded and organized through a worksheet[4]. Here, the effect of the failure mode can be confirmed through the experience of the evaluator, drawings, or simulations. The RPN is used in the evaluation, which indicates the S, O, and D when performing FMEA[5].”
“A molten-carbonate fuel cell (MCFC) generally consists of a regulator, desulfurizer, humidifier, pre-converter, super heater, recycle blower, fresh air blower, inline heater, and catalytic oxidizer[41]. However, MCFCs for ships are comprised of the following components as shown in the block diagram of Figure 7: an air supply system, fuel supply system, water process system, pre-reformer system, fuel cell stack, fresh water system, auxiliary boiler and steam system, and cargo handling system[42].”
Reviewer 2 Report
Dear Authors,
This quality of paper get improved and the most problems are well addressed.
However, the first name are incorrectly used before the comma for the reference 40 and 41. Please make sure the last name is used before the comma for each author in all references.
After you make the corrections, I think the paper can be published.
Thank you.
Author Response
Reviewers' comments:
Reviewer 2
We are very grateful for your line-by-line comments across the manuscript. Your comments are valuable and very helpful for revising and improving our paper, as well as the important guiding significance to our research. We have studied the comments carefully and have revised the manuscript accordingly, which we hope meet with approval.
We put our responses to your comments on the original manuscript as well as the main corrections in the paper and the responses to your official comments are as follows.
All changes were highlighted
This quality of paper get improved and the most problems are well addressed.
However, the first name are incorrectly used before the comma for the reference 40 and 41. Please make sure the last name is used before the comma for each author in all references.
RESPONSE
We highly appreciate your valuable comment. Following your suggestion, we modified reference in the revised manuscript.
“Larminie, J.; Dicks, A. Fuel Cell Systems Explained.; Wiely:New Jersey, USA, 2003; 187-190, 239-241.”
“Zhang, X.; Higier, A.; Zhang, X.; Liu, H. Experimental studies of effect of land width in PEM fuel cells with serpentine
flow field and carbon cloth. Energies. 2019, 12, 471.”
Reviewer 3 Report
The reviewer would like to thank the authors for their polite answer.
But in two cases they have not understood the respective question.
->In lines 189-199 the mathematical background is completely absent.
Now in lines 190-200 there is a light literary description of the mathematical background. In "https://en.wikipedia.org/wiki/Kendall%27s_W" there is a better description. Here the authors should write the respective equations which were used, as well as why they have used theses indexes and not something else, i.e. correlation indexes....
->Similarly in lines 247-249 the Cpk statistical tool should be described completely (especially the mathematical base). Additionally values of Cpk i.e. 1.33 should be justified.
Now in line 255-258 there is an also additional literary description, but the mathematical base is absent (see https://en.wikipedia.org/wiki/Process_capability_index). THe authors should define their parameter and they should explain why Cpk is 1.33 and 2.00 or 1.15.....
Author Response
Reviewers' comments:
Reviewer 3
We are very grateful for your line-by-line comments across the manuscript. Your comments are valuable and very helpful for revising and improving our paper, as well as the important guiding significance to our research. We have studied the comments carefully and have revised the manuscript accordingly, which we hope meet with approval.
We put our responses to your comments on the original manuscript as well as the main corrections in the paper and the responses to your official comments are as follows.
All changes were highlighted
The reviewer would like to thank the authors for their polite answer.
But in two cases they have not understood the respective question.
1) In lines 189-199 the mathematical background is completely absent.
Now in lines 190-200 there is a light literary description of the mathematical background. In "https://en.wikipedia.org/wiki/Kendall%27s_W" there is a better description. Here the authors should write the respective equations which were used, as well as why they have used theses indexes and not something else, i.e. correlation indexes....
RESPONSE
Following your suggestion, we modified Kendall’s coefficients. There are many ways to find correlations, but the most common correlation coefficients are Pearson, Kendall, and Spearman. For the FMEA evaluation items, non-parametric test was applied instead of parametric test because an analysis method that directly calculates the probability and statistically test the data is appropriate regardless of the shape of the population. Pearson is basically used for correlation analysis, but since it is a parametric test that shows correlations when variables are continuous data, one of Kendall and Spearman's methods is used to apply nonparametric tests without linear correlation. Spearman generally has higher values ​​than Kendall's correlation coefficient, but is sensitive to deviations and errors in the data. Therefore, Kendall's correlation coefficient is applied in this study because the sample size is small and the data dynamics are large. In addition, we added mathematical background for kendall’s coefficients.
“There are many ways to find correlations, but the most common correlation coefficients are Pearson, Kendall, and Spearman. For the FMEA evaluation items, non-parametric test was applied instead of parametric test because an analysis method that directly calculates the probability and statistically test the data is appropriate regardless of the shape of the population. Pearson is basically used for correlation analysis, but since it is a parametric test that shows correlations when variables are continuous data, one of Kendall and Spearman's methods is used to apply nonparametric tests without linear correlation. Spearman generally has higher values ​​than Kendall's correlation coefficient, but is sensitive to deviations and errors in the data. Therefore, Kendall's correlation coefficient is applied in this study because the sample size is small and the data dynamics are large.”
“This study calculated Kendall's concordance coefficient using Equation (6) ~ (8) and Statistical Package for the Social Sciences (SPSS), a widely used program in statistical analysis. The coefficient was calculated to analyze the concordance between the evaluators for the reestablished S, O, and D evaluation results.
where is the sum of the classes assigned to each target item by the evaluators, is the number of evaluators, and is the number of target items.
The formula for calculating finds the mean( for the sum of sequence scales.
Then, the average deviation for each item can be obtained as follows.
2) Similarly in lines 247-249 the Cpk statistical tool should be described completely (especially the mathematical base). Additionally values of Cpk i.e. 1.33 should be justified.
Now in line 255-258 there is an also additional literary description, but the mathematical base is absent (see https://en.wikipedia.org/wiki/Process_capability_index). The authors should define their parameter and they should explain why Cpk is 1.33 and 2.00 or 1.15.....
RESPONSE
Thank you for your detailed feedback. The evaluation criteria for incidence, one of FMEA's RPN evaluation factors, were divided into four stages. The three-stage high incidence rate is expressed in PPM, and the PPM index is used to more clearly see how the process curve performs on both sides of the process curve. Where PPM represents one millionth of a unit. As PPM is a more comprehensive capability index than Cpk, in the fourth stage, Cpk was applied to further refine the capability measurement to apply the evaluation criteria. In addition, we added mathematical base for Cpk value.
“Table 3 shows the O criteria, one of the RPN evaluation factors for fuel cell-based hybrid power systems. To precisely evaluate O, the evaluation criteria were classified into 1 (Failure occurrence frequency), 2 (Possibility of occurrence), 3 (High occurrence rate), and 4 (Cpk value). In the third stage, the high incidence rate was evaluated by applying the PPM index and the Cpk statistical tool was used, which measures the ability of the process to produce output within the required specifications. Cpk represents Capability of Process. If both sides have specifications(upper and lower limits) and the center of the distribution does not match the median of both specifications, bias occurs. In order to prepare and evaluate the incidence criteria of the entire system in detail, evaluation criteria were divided into three stages and four stages. In general, the O is considered good when Cpk is 1.33 or greater for a system or a process. The method for obtaining Cpk is as follows[40].
To get the value of Cpk, the capability index Cp is required. Cp is calculated to assess the degree of process capability. Cp can be obtained as equation (1).
Here, USL: Upper Specification Limit, LSL: Lower Specification Limit
The value of Cpk can be calculated from the measured data. If there is only an upper limit of the specification, if there is only a lower limit of the specification, it can be divided into a case where both the upper and lower limits of the specification, the calculation formula is as follows.
Only upper limit of specification :
Only lower limit of specification :
When both upper and lower limit are specified :
Where Cp is th capability index, K is the bias.
K is obtained as follow.
Round 3
Reviewer 1 Report
The requested revisions are done. The novelty/contribution of the paper is clarified.
Authors need to remove Wikipedia articles from their reference list. This is unscientific and unacceptable. They need to cite relevant textbooks or tutorial articles of scientists.